

# Annual Dynamics of Global Land Cover and its Long-term Changes from 1982 to 2015

Han Liu[1], Peng Gong[1,2], Jie Wang[2,3], Nicholas Clinton[4], Yuqi Bai[1], Shunlin Liang[5,6]

[1]Ministry of Education Key Laboratory for Earth System Modeling, Department of Earth System Science, Tsinghua University,

Beijing, 100084, China

[2]AI for Earth Lab, Cross-Strait Institute, Tsinghua University, Beijing, 100084, China

[3]State Key Laboratory of Remote Sensing Science, Institute of Remote Sensing and Digital Earth, Chinese Academy of

Sciences, Beijing, 100101, China

[4]Google LLC, 1600 Amphitheatre Parkway, Mountain View, CA 94043 USA

[5]Department of Geographical Sciences, University of Maryland, College Park, MD 20742 USA

[6]School of Remote Sensing Information Engineering, Wuhan University, Wuhan, 430072, China

*Correspondence to*: Peng Gong (penggong@tsinghua.edu.cn), Jie Wang (sohuwangjie@163.com)

**Abstract.** Land cover (LC) is *an* important terrestrial variable and key information for understanding the interaction between

human activities and global change. As the cause and result of global environmental change, land cover change (LCC)

influences the global energy balance and biogeochemical cycles. Continuous and dynamic monitoring of global LC is urgently

needed. Effective monitoring and comprehensive analysis of LCC at the global scale is rare. Using the latest version of GLASS

(The Global Land Surface Satellite) CDRs (Climate Data Records) from 1982 to 2015, we built the first set of CDRs to record

the annual dynamics of global land cover (GLASS-GLC) at 5 km resolution using the Google Earth Engine (GEE) platform.

Compared to earlier global LC products, GLASS-GLC is characterized by high consistency, more detailed classes, and longer

temporal coverage. The average overall accuracy is 85 %. We implemented a systematic uncertainty analysis at the global

scale. In addition, we carried out a comprehensive spatiotemporal pattern analysis. Significant changes and patterns at various

scales were found, including deforestation and agricultural land expansion in the tropics, afforestation and forest expansion in

northern high latitudes, land degradation in Asian grassland and reclamation in northeast China, etc. A global quantitative

analysis of human factors showed that the average human impact level in areas with significant LCC was about 25.49 %. The

anthropogenic influence has a strong correlation with the noticeable Earth greening. Based on GLASS-GLC, we can conduct

long-term LCC analysis, improve our understanding of global environmental change, and mitigate its negative impact.

GLASS-GLC will be further applied in Earth system modeling in order to facilitate research on global carbon and water cycling,

vegetation dynamics and climate change. The data set presented in this article is published in the Tagged Image File Format

(TIFF) at https://doi.org/10.1594/PANGAEA.898096. The data set includes 34 TIFF files and one instruction doc file.



## 1 Introduction

Land cover (LC) is the physical evidence on Earth. It is the result of both natural and human forces (Running, 2008;Sterling

et al., 2013;Tucker et al., 1985;Gong et al., 2013;Yang et al., 2013). It is an important source of information to understand the

complex interaction between human activities and global changes (Lambin et al., 2006). LC data is one of the most important

variables needed to bring about the nine large social benefits in the field of Global Earth Observation Systems (Herold et al.,

2008). Land cover change (LCC) is the cause and result of global environmental change (Turner et al., 2007), and it can change

the energy balance and biogeochemical cycles (DeFries et al., 1999;Claussen et al., 2001), further affecting climate change

and surface attributes and the provision of ecosystem services (Pielke, 2005;Zhao et al., 2001;Gibbard et al., 2005;Reyers et

al., 2009). Therefore, a long time series of LCC information is critical to the understanding of global environmental change

(Matthews et al., 2004). LC and LCC information is also valuable to resource management, biodiversity conservation, food

security, forest carbon, etc (Houghton et al., 2012;Achard et al., 2004;Andrew K et al., 2015). Therefore, more frequent land

cover information at the global scale is highly desirable.

However, LC is highly dynamic due to changes in natural phenology and human activities (Lambin et al., 2001). This

characteristic poses a huge challenge to mapping and monitoring (Verburg et al., 2009;Lepers et al., 2005;Rindfuss et al., 2004),

and an effective quantitative analysis of global LCC is lacking (Ramankutty et al., 2006). The traditional method of LC

mapping based on field studies can hardly be applied to large areas due to the required amount of labor (Gong, 2012). In

addition, any mapping results obtained in this way would be difficult to update in a timely manner. Satellite observations are

the most economical and feasible means of large-scale LC monitoring (Fuller et al., 2003;Rogan and Chen, 2004). Due to the

development of satellite sensors, the continuous accumulation of historical satellite data, and the advancement of relevant

image processing algorithms, LC monitoring can be effectively carried out (Cihlar, 2000;Pal, 2005;Gallego, 2004;Chen et al.,

2018). However, previous monitoring mainly focuses on the mapping of a particular area (Liu et al., 2002;Brink and Eva,

2009;Yuan et al., 2005;Margono et al., 2012;Feng et al., 2018) or in a single period (Homer et al., 2004), and because of the

differences in data sources and mapping methods, the consistency of mapping results between different sources and periods is

poor and lacks comparability, making it difficult to quantify the changes effectively (Friedl et al., 2010).

Automatic mapping methods depend highly on the sample dataset for its representativeness, quantity and quality due to the

considerable heterogeneity at the global level (Gong et al., 2013;Li et al., 2014). A combination of a comprehensive global

sample dataset, professional interpretation and support from mapping teams are needed (Li et al., 2017). In general, sample

LC data are mainly collected from field visits or manual interpretation (Li et al., 2016;Hansen et al., 2000). Generalization

from higher resolution LC map products can also be useful for coarser resolution mapping purposes (Song et al., 2018a). The

former is more accurate and effective, but requires much manpower, resource and effort (Li et al., 2016); the latter is a feasible

option and is more efficient but largely depending on the accuracy of the parent product.



A number of global LC products exists. Some examples include the 30 m Finer resolution observation and monitoring of global land cover (FROM-GLC) (Gong et al., 2013), the 1992-2015 annual 300 m global land cover data (http://maps.elie.ucl.ac.be/CCI/viewer/index.php), MODIS global land cover product (Friedl et al., 2010), 1 km International Geosphere-Biosphere Programme Data and Information System Cover map (IGBP-DISCover) (Loveland et al., 2000), 1 km

University of Maryland (UMD) land-cover map (Hansen et al., 2000), 1 km Global Land Cover 2000 (GLC2000) map (Bartholome and Belward, 2005). These mapping results tend to focus on a single or short period of time, and because of their different classification systems and resolutions, they are difficult to compare (Ban et al., 2015;Grekousis et al., 2015). However, high-resolution mapping results can be used as an effective reference for low-resolution mapping (Song et al., 2018a;DeFries et al., 1998). Therefore, when performing lower-resolution global mapping, it is possible to consider directly generating

training samples from high-resolution global mapping results, which will meet the mapping requirements, (Wang et al., 2016). Long time-series LC mapping requires high consistency of data sources, and also has certain requirements for multi-period samples (Wardlow and Egbert, 2008). The commonly used satellite data that cover a long period of time (more than 30 years) include the Advanced Very High Resolution Radiometer (AVHRR) data and Landsat imagery (Giri et al., 2013;Franch et al., 2017;Wulder et al., 2008). While Landsat data has higher resolution, in many areas they are more prone to cloud, consistency

and data volume (Gómez et al., 2016;Wulder et al., 2008;Xie et al., 2018). AVHRR data has a low spatial resolution, and the quality of the raw AVHRR data is poor. The requirements for pre-processing and consistency processing such as cloud removal and missing value filling are high. The GLASS CDRs based on AVHRR data tend to have better data consistency due to the systematic data production (Liang et al., 2013). Using such data for LC mapping can significantly improve the consistency and comparability of mapping results, and thus can be effective in supporting change analysis. If the consistency of the original

data source used is not good enough, it may be necessary to collect annual samples for classification to ensure the reliability of change analysis (Xu et al., 2018).

Recently, some attempts have been made to map global LC over a long time series, but these have focused on a single class (such as water bodies (Wood et al., 2011;Pekel et al., 2016), impervious surface (Schneider et al., 2010;Zhang and Seto, 2011), cropland (Pittman et al., 2010), etc.) or a few classes (such as Vegetation Continuous Fields (VCF) (Song et al., 2018a), mainly

depicting vegetation changes). General purpose multi-class land cover mapping over a period of over 30 years does not exist before.

Because of the lack of long time-series general purpose global LC maps, using the Google Earth Engine (GEE) platform (Gorelick et al., 2017), we produced the first CDR set of consistent and reliable LC products, GLASS-GLC, covering the period from 1982 to 2015. The data used was primarily the 0.05 ° AVHRR-based GLASS CDRs. The classification system is

adjusted from the FROM-GLC according to the data characteristics. Below, we describe the methods used, results obtained with some preliminary change analysis.



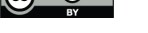

## 2 Data and methods

The framework for mapping GLASS-GLC is shown in Fig. 1. It includes annual feature collection and construction, training

sample generation, classification and time consistency adjustment, accuracy assessment and product inter-comparison. The

entire framework is implemented in the GEE. The GEE is a cloud-based platform for planetary-scale geospatial analysis that

brings Google's massive computational capabilities to bear on a variety of high-impact societal issues including deforestation,

drought, disaster, disease, food security, water management, climate monitoring and environmental protection (Gorelick et al.,

2017). We uploaded GLASS data to GEE and did subsequent analysis in GEE.

### 2.1 Data

The annual feature collection from 1982 to 2015 involves a variety of data products, the most important of which is the latest

version of GLASS CDRs. CDRs require data with a long time series, high consistency and high continuity, which is not the

same as the commonly-used remote sensing products (Hollmann et al., 2013;Cao et al., 2008). Derived from AVHRR data, the

GLASS CDRs include a wide range of surface parameters that are important for LC classification (http://glass-

product.bnu.edu.cn/). The products have a spatial resolution of 0.05 °, a temporal frequency of 8 days with a time span of

1982-2015. In our study, Normalized Difference Vegetation Index (NDVI), Leaf Area Index (LAI) (Xiao et al., 2016), Fraction

of Absorbed Photosynthetically Active Radiation (FAPAR) (Xiao et al., 2015), Evapotranspiration (ET) (Yao et al., 2014),

Gross Primary Production (GPP) (Yuan et al., 2010), Broadband Emissivity (BBE) (Cheng et al., 2016), White-sky Albedo in

Visible band (ABD_WSA_VIS), White-sky Albedo in Near Infrared band (ABD_BSA_NIR) and White-sky Albedo in

Shortwave band (ABD_WSA_shortwave) (Qu et al., 2014) are the variables used for subsequent classification.

To provide further reference, vegetation cover fraction (VCF) products are used to aid classification. The VCF products express

the surface as a combination of vegetation proportions according to information from remotely sensed data. To match the

resolution of the GLASS CDRs, the VCF products used here (Song et al., 2018a) also have a spatial resolution of 0.05 °, and

are obtained from the Land Processes Distributed Active Archive Center (https://lpdaac.usgs.gov/). These products are mainly

based on AVHRR, and the interannual consistency has been maintained. Based on the training samples from Landsat products

from around 2000 (Hansen et al., 2013;Ying et al., 2017), with a supervised regression tree model, the VCF products from

1982 to 2016 (data missing in 1994 and 2000) were generated, and were composed of the percentages of tree canopy (TC),

short vegetation (SV) and bare ground (BG) in each pixel.

In addition, in order to enhance the distinguishing capacity, we also used terrain data provided by the Global Multi-resolution

Terrain Elevation Data of 2010 (GMTED2010). Based on the elevation data, the slope information can be further calculated

to reflect the terrain and help to distinguish different vegetation types growing on steep slopes to those on level ground. The

dataset comes from the GEE platform and contains 2010 Earth Elevation data collected from various sources. The primary

source is the Shuttle Radar Topography Mission (SRTM) Digital Terrain Elevation Data (DTED) (void-filled) 1-arc-second

data. Other sources are used for filling the gaps in areas outside the SRTM coverage. As the terrain is relatively stable over

years, using the data of one single year is plausible. The spatial resolution of the GMTED2010 data used is 7.5 arc seconds

and it has been upsampled to 5 km in subsequent analyses.

**2.2 Classification system**

The classification system in FROM-GLC Version 2 (FROM-GLC_v2) defines eleven Level 1 classes that can be easily mapped

to the Food and Agricultural Organization of the United Nations (FAO) LC Classification System and the International

Geosphere–Biosphere Programme (IGBP) classification system (Wang et al., 2015). This classification system evolved from

the classification system of FROM-GLC Version 1 (Gong et al., 2013) with addition of leaf information.

We adjusted some classes of the original classification system according to the spatial resolution and situation of the used data.

Because the data used here are land surface products, where the water surface has been masked, the class of "water bodies"

cannot be extracted from the GLASS dataset. Wetland is a highly variable class and impervious surface whose patches are

small in size. They are difficult to identify at the spatial resolution of 0.05 °(Wang et al., 2015). Thus, the water body,

impervious surface, and wetland classes were not included in this work, and they shall be derived with more specialized

methods. While water and impervious surface mapping have achieved satisfactory results (Ji et al.;Gong et al., submitted),

wetland mapping remains a great challenge (Gong et al., 2013). In addition, the "cloud" class was removed. The adjusted

classification system consists of 7 classes, including cropland, forest, grassland, shrubland, tundra, barren land, snow/ice, as

shown in Table 1.

**2.3 Training samples**

In order to obtain the training samples, we adopted the majority-class synthesis strategy. First, we projected the 30m FROM-

GLC_v2 results, that were created using Landsat data acquired mainly from 2013-2015 (Li et al., 2017), into a 0.05 ° coordinate

system. By calculating the area ratio of each class in each 0.05 ° pixel, the class with the greatest area ratio in each pixel was

used as the new class label in the aggregated 0.05 ° mapping results. Subsequently, sample points were randomly generated

(with a limited interval greater than 0.1 °) with the class label obtained from the aggregated FROM-GLC_v2 0.05 ° mapping

result (adjusted to be consistent with the new classification system). Finally, 10,000 training sample units were obtained. The

spatial distribution of training sample units is shown in Fig. 2, and the class distribution of training samples is shown in the

inner pie chart.

**2.4 Feature collection**

We constructed a feature collection with a strong discrimination ability to detect LC from multiple aspects such as terrain,

phenology, spectrum, and spectral index, etc. The annual percentiles (including 0, 10, 25, 50, 75, 90, 100) of all bands of the

GLASS CDRs and the mean and standard deviation of the NDVI between two adjacent percentiles are calculated, as an annual feature collection from GLASS CDRs. Among them, the percentile that represents specific phenological information can provide simplified time series information, reduce the noise of annual time series, and help improve the classification accuracy

(Hansen et al., 2013). By extracting the statistical information between adjacent percentiles, the time series information can be further supplemented. Due to the systematic deviation of AVHRR products (Song et al., 2018b), in order to ensure the inter-annual consistency of the GLASS features, we used the processing method developed for generating the VCF products, with the corresponding MODIS products for end-member correction, where desert and intact forest are regarded as the end-element of each pixel (Song et al., 2018a). After the correction, the inter-annual inconsistency of feature collection from the GLASS

CDRs is improved. Figure 3 shows the time series of the global median value of the GLASS ABD_WSA_VIS band, where the orange one represents the curve before the correction and the grey one is the result after the correction. It can be seen that after the correction, the fluctuations of the feature become smaller, and the individual abnormal values are also adjusted.

Taking into account the time span of the GLASS CDR-based feature collection, the VCF products from 1982 to 2015 are used, with the missing 1994 and 2000 data supplemented by calculating the average of the adjacent years. There are three features

of the percentage of tree cover (TC), short vegetation (SV) and bare ground (BG) for each year. Based on the GMTED2010 dataset, the slope information is calculated and finally added to obtain an average slope value for each 0.05 ° pixel. In addition, the central latitude and longitude information of each 0.05 ° pixel is also recorded as part of the input features. Finally, an annual collection of 81 input features for the period of 1982 to 2015 was constructed, including the annual GLASS CDR percentile feature (7×9), the mean and standard deviation of the NDVI annual adjacent percentiles (6×2) and VCF features (3),

assisting the slope information (1) and latitude (1), longitude (1) information (Table 2).

**2.5 Classification and time consistency**

We used a random forest classifier for global LC mapping following the good performance of the random forest classifier in the machine learning field (Rodriguez-Galiano et al., 2012;Pal, 2005). The number of trees was 200, and other parameters were set as default. The classifier was trained using the training sample with an annual feature collection constructed as the

input. The global LC maps from 1982 to 2015 were obtained using the trained classifier.

In order to further ensure the time consistency of the mapping results, we used the "LandTrendr" method (Kennedy et al., 2010;Cohen et al., 2018) and implemented a linear regression-based algorithm for the constructed annual feature collection to find the breakpoints in the time series (Li et al., 2018). The class labels in the time series between adjacent breakpoints will be updated to the mode of the class label time series for the time period. Through this strategy, we can smooth the time series of

the mapping results, avoid noise interference as much as possible, and finally get the adjusted GLASS-GLC.



### 2.6 Accuracy assessment

To verify the reliability of GLASS-GLC CDR products from multiple perspectives, we performed accuracy assessments and uncertainty analyses. Testing samples was extracted from the 30m resolution FROM-GLC_v2 (Li et al., 2017) to evaluate the 2015 LC mapping results. First, we dropped those sample units whose classes were not included in our classification system.

The remaining test samples units were then overlapped with the abovementioned aggregated 0.05 ° FROM-GLC_v2 mapping result, and only those whose class labels were consistent were kept. These were regarded as huge homogeneous samples (H-homo samples) reserved as the final test samples. A total of 23459 huge homogeneous test samples units from FROM-GLC_v2 were obtained to test the 2015 global LC mapping result. In addition, another 525 test samples units from the FLUXNET site data (Gong, 2009) for 2015 were selected to supplement the test samples to further test the 2015 result. The distribution of the

entire test samples in 2015 is shown in Fig. 4, where the class distribution of the test samples is shown in the inner pie chart. In addition to obtaining the classification confusion matrix in 2015 based on the above test samples, in order to identify regions where classification is difficult, an uncertainty analysis was carried out. The incorrect test samples locations are marked as 1, while the correct test samples locations are marked as 0. The spatial distribution map of the uncertainty of the LC mapping result in 2015 is depicted based on a Kriging interpolation method (Oliver and Webster, 1990). The search radius parameter of

Kriging interpolation is set to 12 nearby points, the other parameters as default. The value of the uncertainty ranges from 0 to 1. A value near 0, indicates a lower uncertainty while a value near to 1, indicates a higher uncertainty and a higher possibility of misclassification.

### 2.7 Statistical analysis

To extract the area of LCC, we estimated the trend of change through statistical analysis and avoided the influence of abnormal

fluctuations from the obtained long time series of global LC products. The annual area of each class on the scales of latitudinal zones, continents are summarized. A time series of the annual area for each class was generated. The boundary data of countries and continents were obtained from the Bureau of Surveying and Mapping of China. Eco-region data were obtained from the FAO    global    eco-region    dataset    (Simons    et    al.,    2001) (http://www.fao.org/geonetwork/srv/en/metadata.show?CurrTab=simple&id=1255).

Although the inter-annual consistency has been ensured as much as possible in the above mapping framework, the effects of inter-annual changes due to climate conditions and phenological changes were removed by fitting a linear trend (Theil-Sen estimator (Sen, 1968))   to estimate the long-term trend of change in area for each class, where the annual change slope and the 95 % confidence interval of the slope is given. In addition, a Mann-Kendall test (Mann, 1945) was used to test the trend of time series and the p-value is given. If $p < 0.05$, it is considered that the trend of change is significant.

Further, we got the change mask where all pixels showed a significant change trend guaranteed by statistical hypothesis testing (Wang et al., 2016). First, we downscaled the grid from 0.05 ° to 0.25 °, and the time series of the area ratio of all classes in



each 0.25 ° grid was summed. Using the Mann-Kendall test, those grids showing a significant change (p < 0.05) were obtained. Then the annual change in slope of area ratio for each grid with an increasing or decreasing trend was found using a Theil-Sen estimator. The change ratios were then summarized for the regional scales to estimate the corresponding significant areas of change from 1982 to 2015.

### 2.8 LC conversion

In order to quantify the magnitude of global LCC between 1982 and 2015 and reveal the global temporal LCC pattern, we calculated the ratio of annual global LCC to the global total terrestrial LC area by different time periods. To ensure the quantified LCC to be non-accidental, we limited the computation area within the change mask in which all grids show a statistically significant loss or gain trend. We then summarized the annual LCC by 5-year and 10-year time intervals, respectively.

To further identify the direct causes of LCC, we assessed the LC conversion from 1982 to 2015. Based on the 0.05 ° LC mapping results of 1982 and 2015, a map of LC conversion can be obtained. The computation was also limited to the change mask to ensure the statistical significance. The conversion sources and destinations of LC classes were separately computed, so as to directly assess the direct causes of change in various classes of LC.

### 2.9 Human impact

To further explore the role of human impact in regions with significant LCC, the results are evaluated based on data from the human impact campaign (Fritz et al., 2017), which can be downloaded from https://doi.pangaea.de/10.1594/PANGAEA.869680. The original study area was generated in the 2011 campaign to evaluate a map of land availability for biofuel production (Fritz et al., 2013), collected using a Geo-Wiki crowdsourcing platform. Pixels with a resolution of 1 km were randomly provided to volunteers. For each pixel, volunteers needed to point out the overall degree of human impact (HI, 0-100 %) which was visible from Google Earth's high-resolution satellite image and they were required to provide confidence levels in four categories: unsure; less sure; quite sure; and sure. Here, HI refers to the degree to which the landscape modified by humans visible from satellite images (Fritz et al., 2017). A total of 151942 point-records are available. To get the global distribution map of HI, we performed Kriging interpolation on the point records that had previously excluded the category of unsure confidence level. The search radius parameter of the Kriging interpolation was set to 12 nearby points and the other parameters as default. As shown in Fig. 5, we can see that the interpolation results reflect the global distribution of the intensity of human activity.



## 3 Results

### 3.1 Reliability of the products

The global LC mapping result in 2015 is shown in Fig. 6. Its accuracy was tested with the H-homo sample in 2015 to obtain a

confusion matrix (Table 3). The overall accuracy for the year 2015 reached 86.51 %. As for each class, the accuracies of forest,

barren land and tundra are relatively high, where the user's accuracies and producer's accuracies are over 90 %. The accuracy

of cropland is also high, with the user's accuracy and producer's accuracy reaching 73.54 % and 78.62 %, respectively. The

user's accuracy of shrubland reached 83.62 %, while that of grassland is 67.58 %. Grassland is mainly mixed with cropland

and shrubland. Table 4 shows the testing results of the FLUXNET test samples in which the number of sample units for

shrubland, tundra, barren land, and snow/ice are relatively small. The overall accuracy of all classes is 82 % tested against the

FLUXNET sample. Among them, the user's accuracy and the producer's accuracy for forest reach 91 % and 88 %, respectively.

The producer's accuracy for cropland is 69 %, while its user's accuracy is 73 %.

Putting the test results from FROM-GLC_v2 and FLUXNET together, a spatial distribution map of the uncertainty of the 2015

LC mapping result was generated. As can be seen from Fig. 7, most of the world is shown in a green color, which means that

the mapping result for most regions is most likely to be correct, and the result for 2015 is highly credible. There are still some

regions showing a yellow or orange color, and a smaller number of regions showing red, representing those regions that may

have been misclassified. Since there are no test samples in Greenland., the interpolation results are ignored. In general, the

places with high uncertainty are Africa, East and South America, South Alaska, North and East Australia and Southwest

Indonesia.

### 3.2 Spatiotemporal patterns in LCC

### 3.2.1 Global temporal patterns

Figure 8 shows the variation curves of the global area for various LC classes from 1982 to 2015, where dotted lines are the

corresponding trend lines. Overall, the global area of forest increases significantly (p = 0.0000) from 1982 to 2015. As for

shrubland, although fluctuating, it shows a significant increasing trend (p = 0.0017). The global area of grassland, tundra,

barren land snow/ice significantly decreases with p = 0.0000, p = 0.0019, p = 0.0000, and p = 0.0003 respectively.

Figure 9 shows the ratio of annual global LCC to the global total terrestrial LC area, calculated by different time periods, where

Fig. 9(a) shows the results with a 5-year interval and Fig. 9(b) with a 10-year interval. Overall, the annual ratio ranges from

0.35 % to 0.70 %, with an average of 0.52 % between 1982 and 2015. 5-year interval ratios show a relatively fluctuating trend.

The average ratio reaches 0.63 % in 1991-1995, the highest among the seven intervals. The ratios have relatively large

fluctuations in 2006-2010. All in all, the ratios before 1995 are generally higher, and it gradually decreases since then. With

10-year interval, ratios after 2000 are generally lower with an average of only 0.40 % in 2011-2015.



### 3.2.2 Patterns along latitudinal gradients

The global distribution of 0.25 ° girds with significant LCC from 1982 to 2015 is shown in Fig. 10 for the whole world, where

the color depth represents the estimated change in area ratio per year. The distribution of significant LCC along latitudes is

shown in the right, where the red curve represents a significant increase, green a significant decrease, and blue a net change.

The distribution pattern of LCC along latitudes is different, especially for cropland and forest, where it can be seen that cropland

has increased significantly in the northern tropics and the southern hemisphere. It is confirmed that the significant increase in

cropland has occurred mainly in the tropics and southern hemisphere (Gibbs et al., 2010). Forest has decreased significantly

in the southern hemisphere and has increased significantly in the northern hemisphere, showing regional differences. In

particular, in the high latitudes of the north, forest has increased significantly with a decrease of tundra. However, the increase

in forest area in the northern hemisphere is significantly larger than that in the southern hemisphere, reflecting an overall

increase in total forest area.

The grassland area has reduced at almost all latitudes. This phenomenon may reflect the degradation of grassland. On the other

hand, there might exist an increased trend in global vegetation coverage, where shrubland and forest expansion led to a

reduction in the grassland area. It can be seen that shrubland has increased significantly in the southern hemisphere,

corresponding to the reduction in the grassland area there. The area of barren land is decreasing, especially in the middle and

high latitudes of the north, which further reflects the increase in vegetation coverage. The area of snow/ice in the northern high

latitudes has reduced.

### 3.2.3 Continental patterns

The statistical results for each class at the continental scale are shown in Table 5, Table 6, Table 7, Table 8, Table 9, Table 10

and Table 11, where the slope and p-values are estimated according to the class area time series, while gain and loss are the

computed values from 0.25 ° grids with significant LCC.

There is significant geographical heterogeneity among continents due to differences in latitude and longitude, as well as

economic and social development differences, where significant causes of LCC are from both natural and human influences

(Lambin et al., 2001).

Cropland significantly increased in South America, with a growth rate of $9.1×10^3$ $km^2$/year (p = 0.0108). The area of

significantly increased cropland in Asia and Africa reached $67×10^3$ $km^2$ and $23×10^3$ $km^2$, respectively. Many developing

countries in South America, Asia and Africa have relatively poor economic and social development, rapid population growth

and increasing demand for food (Barbier, 2004). At the same time, the international demand for food has increased, stimulating

the export of crop products and requiring access to new land, which ultimately leads to the expansion of cropland.

Corresponding to the increase in cropland, forest decreased significantly in South America, at a rate of $10.8×10^3$ $km^2$/year (p

= 0.0242). Meanwhile, the area of forest in Africa has significantly decreased by $29×10^3$ $km^2$. In In addition to cropland



expansion, the production of fuelwood and charcoal is also an important driving factor for deforestation (Hosonuma et al., 2012).

The area of forest in Asia has increased at the fastest speed. The area of forest in Europe and North America has also increased significantly. Meanwhile, the tundra area in Asia, Europe and North America decreased significantly by $132 \times 10^3$ km$^2$, $12 \times 10^3$ km$^2$ and $22 \times 10^3$ km$^2$, respectively. The increase of forest in Asia, Europe and North America is related to afforestation projects

and forest restoration policies in some regions (Aide et al., 2013;Pan et al., 2011). On the other hand, the increase of forest and the decrease of tundra in the northern high latitudes may be the result of climate warming which promotes forest growth (Zhu et al., 2016). Many studies have shown that in the past 30 years, a warming climate with rising temperatures and melting ice and snow has promoted vegetation growth i.e. greening in the north (Myneni et al., 1997;Park et al., 2016).

Shrubland has increased significantly in Africa at a rate of $47.4 \times 10^3$ km$^2$/year (p = 0.0030). Shrubland also increased

significantly in Oceania, by an area of $38 \times 10^3$ km$^2$. The main source of shrubland conversion is grassland, which can be regarded as another manifestation of greening, where a warming climate makes vegetation grow more vigorously and plant height increase.

The degradation of grassland in Asia is serious. The area of grassland in Asia decreased significantly by $315 \times 10^3$ km$^2$, which may be due to drought (Dangal et al., 2016;Zhang et al., 2018). At the same time, human activity may also play a significant

role. Another reason is overgrazing that may lead to grassland degradation, and the development of irrigation agriculture that can seriously reduce groundwater levels, which will further aggravate drought (Dubovyk et al.). Barren land in Asia also significantly decreased by $82 \times 10^3$ km$^2$, which may imply the effects of desertification control in some regions. The global snow/ice area has decreased significantly, at a speed of $19.2 \times 10^3$ km$^2$/year (p = 0.0003), reflecting the melting of ice and snow under a warming environment.

**3.3 Characteristics of LC coversion**

Whether LCC is caused by natural or human factors, there is often a significant coupling effect. We attempted to find out some high-frequency LC class conversions for the period 1982 to 2015 (Table 12). In addition, the conversion sources and destinations of each LC class are computed separately, as shown in Fig. 11.

Among land converted to cropland in 2015, grassland was the biggest source, accounting for 67.58 %, which indicated that a

large amount of cropland came from reclamation (Liu et al., 2005). 6.61 % of cropland was converted from forest, showing the process of forest destruction. Among land converted to forest, the proportion of cropland reached 21.74 %, partly due to the fact that abandoned croplands were restored to forest. Barren land and grassland were respectively the large sources of grassland and barren land, reflecting the dynamic transformation between the two classes. Grassland accounted for 35.00 % of the increasing source of barren land, indicating the process of land degradation (Bai et al., 2008).

The most frequent direction of conversion from cropland in 1982 was forest, which reached 78.22 %, reflecting the process of



forest expansion. At the same time, forest was also the main cause of loss of grassland and shrubland, which also confirms the process of forest expansion. The conversion of forest to grassland accounted for 59.04 % of all conversions from forest. The main conversion direction of tundra was forest, reaching 64.60 %, indicating an expansion of forest in the high latitudes of the northern hemisphere.

Overall, the increase of forest accounted for the highest proportion of all conversion processes, reaching 44.17 %, reflecting the phenomenon of forest expansion. The increase of grassland and cropland were second and third highest, reaching 19.79 % and 13.64 %, respectively, showing the phenomenon of cropland and pasture expansion. In addition, the proportions of grassland to shrubland and barren land to grassland were 7.73 % and 5.75 %, respectively. Cropland expansion and surface greening were the main phenomena reflected by the changes in global LC from 1982 to 2015.

**3.4 Human impact**

Figure 12 shows different human impact (HI) levels among different LCC areas. Overall, the average HI level in regions with significant changes in all LC classes is 25.49 %, indicating that human activity has a great impact on LCC (Meyer and Turner, 1992). The highest HI level was found in those regions with significant increases in cropland, reaching an average value of 51.38 %. Meanwhile, the HI level of cropland loss reached 48.02 % while the HI level for forest loss was 26.91 %. In addition,

in any change of natural vegetation, such as forest, grassland and shrubland, the HI level in regions of vegetation loss is higher than that of gain, which indicates that human activity has a destructive effect on natural vegetation, while other factors may promote an increase in natural vegetation (Richardson et al., 2013;Cramer et al., 2001).

The HI levels along continents can be found in Fig. 13. The highest level of HI is found in Europe and lowest in Oceania. The HI in Europe reached 46.86 %, indicating that human activity played a relatively important role in regions with significant

LCC. Asia came second, with an HI level of 32.07 %. In South America and Oceania in the southern hemisphere, the overall HI level in the LCC regions is small.

As shown in Fig. 14, the polar regions and the boreal coniferous forest regions at northern high latitudes with significant LCC have lower HI levels, indicating that LCC in those regions may be more related to natural factors like climate change (Buermann et al., 2014;Macias-Fauria et al., 2012). The level of HI in subtropical regions is high, among which HI levels in

subtropical steppe and subtropical humid forest regions reached 38.23 % and 43.90 %, indicating that the role of LC conversion caused by human activity in subtropical climate areas is significant. In addition, in the temperate steppe regions, the HI level in the regions of significant LCC is also high, reaching 39.87 %, which may be due to intense grazing from agricultural activities (Marlon et al., 2008;Bellwood et al., 2011), resulting in the higher HI level. In the tropics the average HI level in dry forest regions is highest among regions of significant LCC, reaching 34.04 %. Such HI level in this eco-region may be caused

by forest destruction, deforestation, and cropland expansion.



### 3.5 Local hotspots of LCC

Regarding LC, more attention tends to be paid to global and regional LCC. At the local scale, we can further explore the hot spots of LCC and investigate the causes of such change by area. The main regions of LCC hotspots are shown in Fig. 15, where the depth of color represents a significant change.

In the north of Eurasia, forest has increased significantly, and that in Siberia has moved northward to the tundra regions, which is mainly the result of climate warming. The increase in temperature and soil moisture (thawing of the permafrost) has promoted plant growth (Berner et al., 2013).

In northern North America, such as Alaska and the north of Canada, forest has also increased but the extent of the increase is weaker than that in North Eurasia. Studies have shown that this may be related to an insignificant temperature rise in North

America and even a slight cooling trend (Wang et al., 2011). In addition, fire disturbance in northern North America has interrupted forest succession (Alcaraz-Segura et al., 2010) and drought disasters in parts of the United States and Canada have increased tree mortality (Van Mantgem et al., 2009;Peng et al., 2011). These could also be possible reasons for the constant, or even decreasing, forest areas in these regions. In addition to climate warming, the decrease of cropland and increase of forest in the eastern part of the United States are related to forest restoration and management measures (Herrick et al., 2010).

In the Great Plains of Central North America, grassland has decreased and cropland has increased. It has been found that rising gasoline prices and the development of biofuels have led to increasing planting areas of corn and soybean in the United States (Lark et al., 2015;Wright and Wimberly, 2013).

In most countries of South America, croplands have expanded substantially and forests have decreased significantly, especially in the southeastern part of the Amazon rainforest (shown in Fig. 16). This corresponds to the expansion of soybean planting

areas and the development of the cattle ranching industry (Zak et al., 2008;De Sy et al., 2015). In these regions, forest destruction and deforestation owing to human factors overtook the increase of vegetation caused by climate warming.

In Southeast Asia, such as Cambodia, Vietnam, Indonesia and Malaysia, forest has also decreased significantly and cropland has increased. The expansion of cash crops (mainly oil palm) plantations and logging activities in Southeast Asia have led to serious destruction of primitive forests (Wilcove et al., 2013;Miettinen et al., 2011). Natural forest has either been turned into

artificial forest or cut down, resulting in huge loss of biodiversity and increased greenhouse gas emissions (Stibig et al., 2013).

In Africa, forest in the northern part of the Congo Basin has expanded while forest in the southern Miombo forest belt has decreased (Devine et al., 2017). Studies have shown that the increase of forest in the northern part of Africa is related to low population growth, increased carbon dioxide and increased precipitation, while the decrease of forest in the southern part corresponds to high population growth (Brandt et al., 2017). It was also found that the rapid integration of global agricultural

markets in recent decades and subsequent urbanization has caused cropland loss and promoted large-scale tropical deforestation in South America and Southeast Asia (Ordway et al., 2017). Increasing land scarcity and stricter land use

regulations in South America and Southeast Asia may prompt export-oriented commodity crops to be outsourced to sub-Saharan Africa.

In China, forest has increased. In addition to carbon dioxide elevation and climate warming, human activities such as

afforestation and agricultural management (such as agricultural intensification) have also had a great impact on this trend (Piao et al., 2015;Guo and Gong, 2016). In addition, the increase in population and demand for food has led to the expansion of cropland in the Loess Plateau regions of China, while the increase in cropland in Northeast China has come mainly from wasteland reclamation (Liu et al., 2014).

Some grassland in Mongolia and Inner Mongolia of China showed a trend of degradation. Studies have shown overgrazing to

be the main cause of vegetation degradation in this region, while drought and soil erosion have played a secondary role (Yin et al., 2018). The obvious increase of grassland areas in the eastern part of the Qinghai-Tibet Plateau implies that the temperature rise has promoted vegetation growth in highly elevated regions, as vegetation growth in this region is usually limited by low temperatures (Wang et al., 2012). The decrease of grassland in central Asia and parts of Western Asia may be related to climate change where the area of land desertification has increased under the influence of drought (Cook et al., 2010).

In some parts of the former Soviet Union in Eastern Europe, a decrease of cropland and an increase of forest can be observed. Studies have found that a large number of cropland areas were abandoned after the dissolution of the Soviet Union and the transition from a planned economy to a market economy in these regions (Meyfroidt et al., 2016;Wertebach et al., 2017;Kuemmerle et al., 2011), which reflects the role of socio-economic systems in LCC.

## 4 Discussions

Based on the accuracy assessment results, it can be seen that the global LC mapping products of 1982-2015, GLASS-GLC are reliable with high accuracies, and the global long-term mapping framework we designed is effective. Using GLASS-GLC CDRs in change analysis of LC can reflect a 34-year global landscape change pattern. Many phenomena and patterns can be confirmed by existing research, such as the expansion of tropical agricultural land, greening in the northern region, deforestation in the southern hemisphere and melting of snow and ice. In addition, we have assessed the impact of human

effects within different LC classes, and have further explored the causes of LCC in local hotspots, combined with field visits and literature reviews.

However, there are still deficiencies in the design of the mapping framework. First, the large grid size of 0.05 °, due to the coarse spatial resolution, can only reflect the average change state of LC in a large area, thus many small-area phenomena cannot be well reflected (Gómez et al., 2016). For example, the reduction of much agricultural land is due to urbanization, and

the expansion of cities is usually sporadic. Although those changes are large at the global scale, they can hardly be reflected with 0.05 ° pixels. Moreover, due to the synthesis principle, the classification result of each pixel can only represent the class



with the largest proportion in area, and the information of remaining classes is ignored even though they can sometimes be more than 50 % in total. Such a neglect, due to the famous "Scale Effect" (Turner et al., 1989) can also cause great deviations in the final statistical summary of the LC area leading to uncertainties when compared with mapping results at finer resolutions.

Second, our sampling strategy for training has certain limitations. On one hand, since the training sample is generated from 30m global mapping results of more than 75 % accuracy, this will inevitably propagate and accumulate error to 5 km resolution. Of course, due to the higher signal-to-noise ratio of the high-resolution data, the sampling is still satisfactory compared to direct visual interpretation of the coarse resolution images. On the other hand, the training sample used is only from a single year of circa 2015. Although we have implemented a time series correction for the original input features and performed a

time-consistent post processing on the classification results, the effects of inter-annual fluctuations of the features cannot be completely avoided (Song et al., 2018a). On the other hand, according to the stable classification with limited sample theory (Gong et al., submitted), a representative sample collected in one year with less than 20 % in error should suffice in multiannual use at the global scale. Therefore, a multi-year sample set may not be as critical for multiannual classification provided the sample is better than 80 % accurate. In our case, although the source training data has an accuracy of 77 % (Gong et al., 2017),

we are not certain if the aggregated sample set exceeds an accuracy of 80 %. This needs further assessment.

For the generation of test samples, we have actually adopted the scale-up approach. That is to say, we first upscaled the 30m test samples set to 5 km by maximum area synthesis, which contains unavoidable errors because of scale transformation. The best way to verify the accuracy, of course, is to use a 0.05 ° test samples set directly derived at this resolution. However, due to the difficulty of visual interpretation in coarse scale and field investigation (Gong et al., 2013), establishing a sample library

at 5 km resolution is not easy. Thus, instead, we adopted the method of aggregation of 30m FROM-GLC_v2 results to 5 km scale to generate samples. It is plausible to regard the selected test samples as "H-homo samples" that can be used for coarse resolution mapping. Although this method is feasible to a certain extent, there are inevitably errors.

We have eliminated wetland and impervious surface in our classification system. This is a tradeoff when working at the 5 km scale. Patches of wetland and impervious surface are usually small, and it is difficult to achieve a pixel size of 0.05 ° for many

situations, so the classification of the two types is extremely difficult. However, both are important LC types. Wetland is a transitional zone between terrestrial ecosystems and aquatic ecosystems (Davidson, 2014). The impervious surface can represent the urban area. In recent years, urban expansion has been a relatively significant phenomenon in global environmental change (Seto et al., 2011). Urban expansion reflects an important type of human activity, so the impervious surface is also one of the essential components to reflect anthropogenic influence though the total area of its change is usually small.

It should be pointed out that at a coarse resolution of 0.05 °, our definition of forest is more inclined to the tree canopy cover. Thus the changes in internal density of trees can also be reflected in the area change of forest, instead of just the stand-replacement type (Korhonen et al., 2006). In addition, the largest-class synthesis strategy we adopted also makes it unavoidable to include internal density change of various class, which in turn will further affect the classification and change area

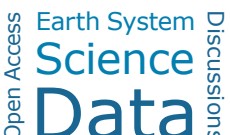

calculation of forest class.

The result of our statistical summary shows that the global vegetated area increased significantly between 1982 and 2015, which is inconsistent with the results of FAO and some other global mapping products. This inconsistency originates from, on one hand, the above limitations of our designed mapping framework, and on the other hand, the statistics collected by FAO, and other census-based datasets, which are also affected by errors from many aspects, and its effectiveness is yet to be evaluated. Nevertheless, many studies can also confirm our results, such as (Song and Hansen, 2017;Song et al., 2018a;Piao et al.,

2015;Pan et al., 2018) who have proved that the global vegetation area has increased with the increasing NDVI and LAI in time. To some extent, the sum of the global change area of forest, shrubland, and grassland is showing an increasing trend in our results, which can be seen as the sign of global vegetation growth.

In addition, because we are mainly depicting the natural biophysical properties of vegetated areas with limitation in resolution, some artificial features cannot be distinguished, such as plantations (rubber, oil palm, and various fruit trees) and natural forest,

which are uniformly included as forest in our classification system.

In the statistical analysis, although we have already conducted post-classification time-consistency processing for the original LC mapping results as much as possible, it is inevitable that there are still large fluctuations and interferences from various unknown factors unfavorable to the extraction of long-term trend of LCC. In order to ensure that the trend of the resulting time series is significant, we have to scale up the classification result from 0.05 ° to 0.25 °, converting the original class label of

each 0.05 ° pixel to the class area ratio of 0.25 ° grid. The long-term time series of the area ratios is tested for statistical significance. However, in some cases this procedure will also be influenced by the "Scale Effect".

In the analysis of anthropogenic influences, indirect effects of many human activities were ignored because the main objective was to include the effects of directly visible human activities. For example, human activities increase the concentration of carbon dioxide in the atmosphere, which in turn affects the global climate, leading to higher temperature, and thus increasing

vegetation coverage (Piao et al., 2006;Bonan, 2008). This pathway of action is indirect, but it is difficult to reflect in the human impact data we use, which results in an underestimation of the assessment of anthropogenic influences.

GLASS-GLCs contain more detailed LC classes, longer temporal coverage (34 years), high consistency, which meet the requirement for CDR. GLASS-GLC CDRs are the first collection of global LC dynamics of 5 km, and fill the existing gap for high-reliability and consistency of long-term general purpose global LC products. In addition, our strategy of generating

samples from high-resolution classification products can greatly reduce the cost and investment of sample collection, and can flexibly and effectively be extended to other coarse-resolution LC mapping tasks in the future.

In the future, with the advancement of technology and the accumulation of remote sensing datasets, the use of remote sensing products for LC mapping with higher resolution and longer time series will undoubtedly better reflect the global LC and its changes. However, under limited conditions, we can consider using coarse-resolution satellite data to determine the locations

of potential rapid change, and then use high-resolution data in these hotspots to accurately estimate the rate and mode of change.

Moreover, it is necessary to establish a multi-year sample library to assess the impact of inter-annual fluctuations in features on the accuracy of change characterization and analysis. Wetland and impervious surface are LC classes that have extremely high value. It would be useful to supplement the mapping and change analysis of these two classes when suitable data become available. For the analysis of global LCC, systematic and in-depth attribution analysis and research can be further carried out.

In addition, the development of LC ratio mapping products (similar to VCF products) with techniques of soft classification, rather than hard classification, especially for the case of coarse resolution, should be considered.

**5 Data availability**

GLASS-GLC products at 5 km resolution from 1982 to 2015 are available to the public in the TIFF format at https://doi.org/10.1594/PANGAEA.898096 (Liu et al., 2019).

GLASS CDRs were provided by Beijing Normal University Data Center (http://glass-product.bnu.edu.cn/, last access: 27 December 2018). VCF products were obtained from the Land Processes Distributed Active Archive Center (https://lpdaac.usgs.gov/, last access: 20 December 2018). GMTED2010 were acquired from Google Earth Engine (https://code.earthengine.google.com/, last access: 24 December 2018). Geo-Wiki points came from the human impact campaign (https://doi.pangaea.de/10.1594/PANGAEA.869680, last access: 30 November 2018). Eco-region data were

obtained           from           the           FAO           global           eco-region           dataset (http://www.fao.org/geonetwork/srv/en/metadata.show?CurrTab=simple&id=1255, last access: 3 December 2018).

**6 Conclusions**

In order to better reflect the global land changes, continuous and dynamic monitoring of global LC is necessary. We built GLASS-GLC, the first CDRs for global LC on the GEE platform. It can capture the global LCC information from 1982 to

2015. Compared to previous global LC products, GLASS-GLC products cover a longer time period and have higher consistency and more detailed classes. Our entire mapping framework is based on FROM-GLC_v2, including the classification system and high-quality H-homo sample generation.

Based on over ten thousand independent test samples units from both the FROM-GLC sample set and FLUXNET site data, the average overall accuracy of GLASS-GLC was shown to exceed 80 %. Using inter-comparisons with other global LC

products of different resolutions from various data sources, we verified the effectiveness and reliability of GLASS-GLC from different perspectives. Systematic uncertainty analysis was also performed on a global scale based on the results of the accuracy assessment and its geographical distribution. This shows that GLASS-GLC CDR products have relatively low uncertainty in most parts of the world. Our results also indicate that GLASS CDRs have potential for multi-class LC mapping and can provide more than enough features and information to distinguish different LC classes, with relatively strong temporal and spatial

consistency, which can produce extremely reliable change information.

Comprehensive spatiotemporal pattern analysis based on GLASS-GLC reflected and revealed many significant global LCC phenomena and patterns, such as deforestation and agricultural land expansion in the tropics, afforestation and forest expansion in the northern regions, etc. An analysis of the global LC conversion pattern from 1982 to 2015 revealed hot spots of LCC such as land degradation, forest restoration, reclamation and agricultural land abandonment.

Since anthropogenic influence has become one of the most important driving forces for LCC, especially after the industrial revolution, we quantified the level of human impact in areas of significant LCC. The results show that the average human impact level in areas of significant LCC are about 25.49 %, suggesting that anthropogenic influence plays a strong role in vegetation destruction, expansion of tropical agricultural land, and degradation of grassland areas, etc. Under the current global climate change scenario with significantly elevated GHG concentrations and temperature rises, this remarkable human impact

has also contributed to a noticeable greening trend of the Earth because of the effect of carbon dioxide fertilization.

Combined with field visits and literature reviews on local LCC hot spots, we can see that global LC is affected by the synergetic effect of many complicated and multi-faceted factors, including human activity, climate change, socio-economic policies, and the natural environment transition, etc., and such change could further influence global and regional climate, environment, biodiversity, etc.

With increasing economic globalization, LCC has increased. Based on GLASS-GLC, effective global LC and change analysis could be conducted, enhancing our understanding of global environmental change, and even mitigating its negative impact to some extent, which is also beneficial to the achievement of sustainable development goals.

**Author contribution**

PG conceived the research. HL and JW designed the experiments and HL carried out experiments. NC provided GEE support.

SL provided data. HL prepared the manuscript with contributions from all co-authors.

**Competing interests**

The authors declare that they have no conflict of interest.

**Acknowledgements**

This work is partially supported by the National Key Research and Development Program of China (NO.2016YFA0600103),

a donation made by Delos Living LLC, and the Cyrus Tang Foundation.



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





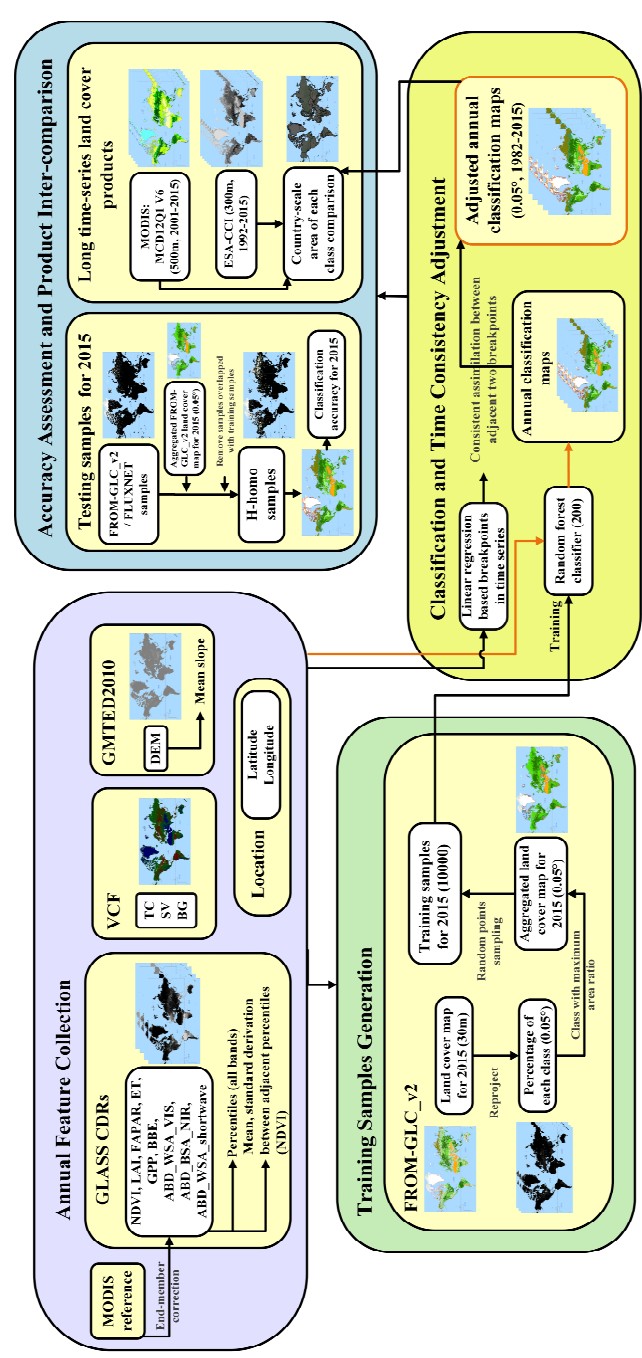

Figure 1: The framework for building GLASS-GLC (annual dynamics of global land cover) CDRs (Climate Data Records).



**Table 1: Classification system, with 7 Level 1 classes and 21 Level 2 classes.**

| Level 1 class | Level 2 class | | | | |
|---|---|---|---|---|---|
| Cropland | Rice fields | Greenhouse farming | Other croplands | Orchards | Bare farmlands |
| Forest | Broadleaf forests, leaf-on | Broadleaf forests, leaf-off | Needleleaf forests, leaf-on | Needleleaf forests, leaf-off | Mixed forests, leaf-on | Mixed forests, leaf-off |
| Grassland | Pastures | Natural grasslands | Grasslands, leaf-off | | |
| Shrubland | Shrublands, leaf-on | Shrublands, leaf-off | | | |
| Tundra | Herbaceous tundra | Shrub and brush tundra | | | |
| Barren land | | | | | |
| Snow/ice | Snow | Ice | | | |





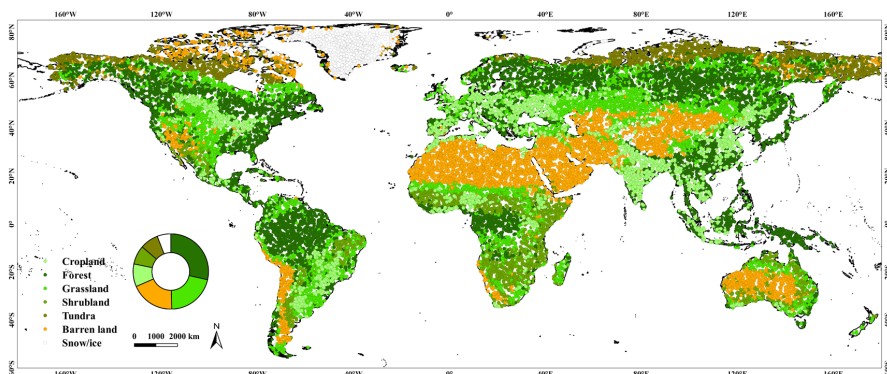

**Figure 2: The geographical distribution of training samples, where different colors represent the different years.**






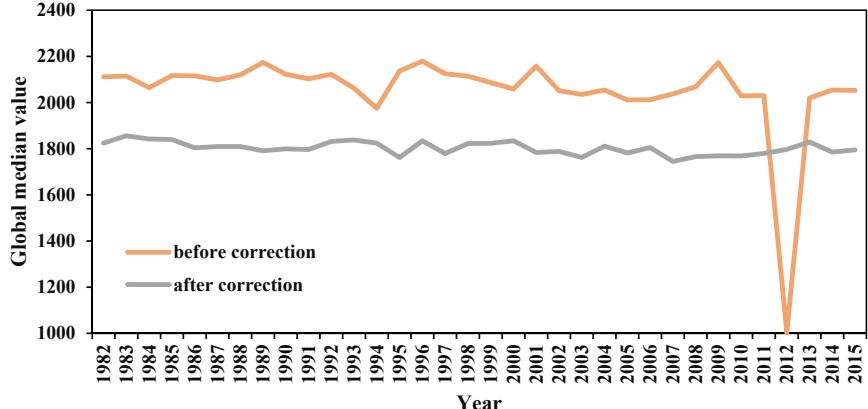

**Figure 3: Global median value time series of GLASS ABD_WSA_VIS before and after the end-member correction with reference to MODIS.**



**Table 2: The explanatory table of the constructed feature collection, with a total 81 features each year.**

| Product | Band | Feature | Number of features |
|---|---|---|---|
| GLASS CDR, 0.05 °, 1982-2015 | NDVI | | |
| | LAI | Percentiles [0, 10, 25, 50, 75, 90, 100] of all 10 bands | 63 |
| | FAPAR | | |
| | ET | | |
| | GPP | | |
| | BBE | Mean, standard derivation of NDVI between adjacent two percentiles of NDVI | 12 |
| | ABD_WSA_VIS | | |
| | ABD_BSA_NIR | | |
| | ABD_WSA_shortwave | | |
| VCF, 0.05 °, 1982-2015 | TC | TC | |
| | SV | SV | 3 |
| | BG | BG | |
| GMTED2010, 7.5 s, 2010 | DEM | Mean slope in each 0.05 ° pixel | 1 |
| Location | Latitude, longitude | Center latitude, longitude of each 0.05 ° pixel | 2 |
| Total number of features | | | 81 |




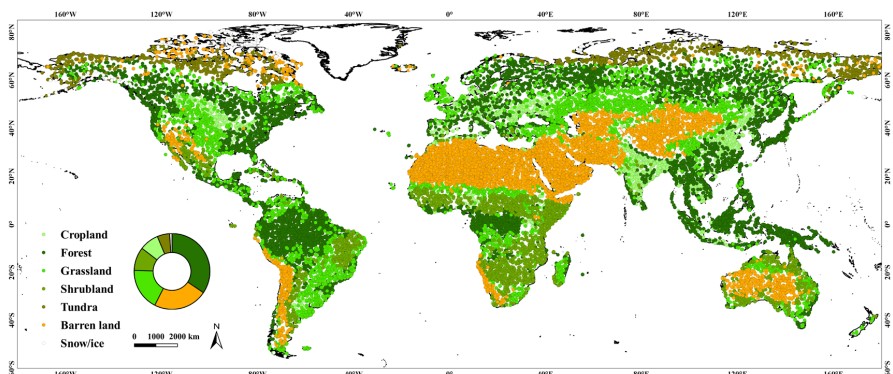

**Figure 4: The geographical distribution of different types of huge homogeneous test samples (H-homo sample), where the different**

**colors represent the source of the sample units, either FROM-GLC_v2 or FLUXNET.**



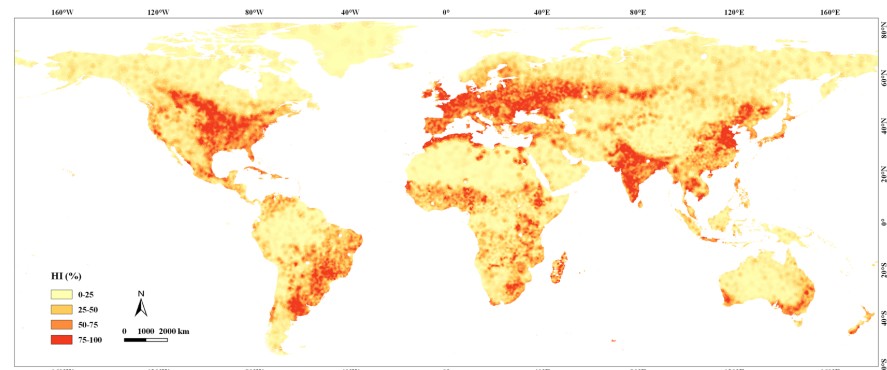

**Figure 5: The geographical distribution of the spatial interpolation results of human impact where the darker color indicates a value closer to 100 and a higher human impact.**




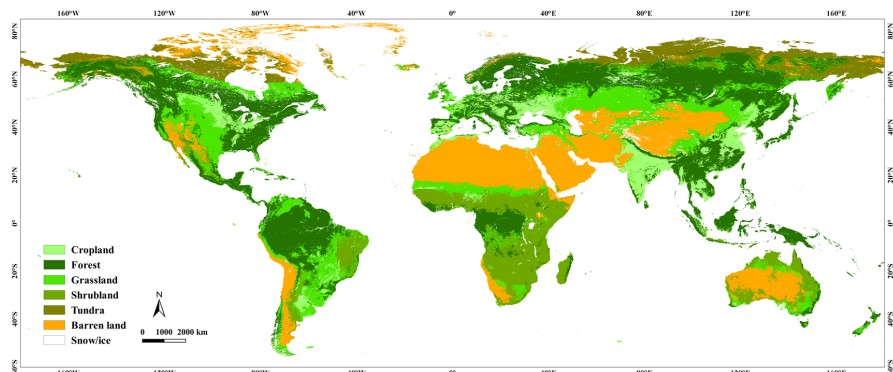

**Figure 6: GLASS-GLC (annual dynamics of global land cover) CDRs (Climate Data Records) result in 2015.**



**Table 3: Classification accuracy in 2015 based on FROM-GLC_v2 test samples. (Overall accuracy = 86.51 %, UA = User's Accuracy**

**and PA = Producer's Accuracy)**

| Class | Cropland | Forest | Grassland | Shrubland | Tundra | Barren land | Snow/ice | Total | UA |
|---|---|---|---|---|---|---|---|---|---|
| Cropland | **1390** | 166 | 221 | 101 | 0 | 12 | 0 | 1890 | 73.54 % |
| Forest | 115 | **7427** | 279 | 145 | 18 | 0 | 3 | 7987 | 92.99 % |
| Grassland | 199 | 431 | **2820** | 534 | 45 | 141 | 3 | 4173 | 67.58 % |
| Shrubland | 47 | 65 | 185 | **1986** | 0 | 92 | 0 | 2375 | 83.62 % |
| Tundra | 0 | 32 | 36 | 0 | **1157** | 24 | 2 | 1251 | 92.49 % |
| Barren land | 17 | 5 | 91 | 27 | 48 | **5336** | 20 | 5544 | 96.25 % |
| Snow/ice | 0 | 2 | 10 | 0 | 7 | 41 | **179** | 239 | 74.90 % |
| Total | 1768 | 8128 | 3642 | 2793 | 1275 | 5646 | 207 | **23459** | |
| PA | 78.62 % | 91.38 % | 77.43 % | 71.11 % | 90.75 % | 94.51 % | 86.47 % | | **86.51 %** |



**Table 4: Classification accuracy in 2015 based on FLUXNET testing samples. (Overall accuracy = 82.10 %, UA = User's Accuracy**

**and PA = Producer's Accuracy)**

| Class | Cropland | Forest | Grassland | Shrubland | Tundra | Barren land | Snow/ice | Total number | UA |
|---|---|---|---|---|---|---|---|---|---|
| Cropland | **63** | 5 | 17 | 1 | 0 | 0 | 0 | 86 | 73.26 % |
| Forest | 13 | **243** | 9 | 2 | 0 | 0 | 0 | 267 | 91.01 % |
| Grassland | 8 | 21 | **91** | 2 | 0 | 2 | 0 | 124 | 73.39 % |
| Shrubland | 7 | 3 | 0 | **19** | 0 | 0 | 0 | 29 | 65.52 % |
| Tundra | 0 | 3 | 0 | 0 | **14** | 0 | 0 | 17 | 82.35 % |
| Barren land | 0 | 1 | 0 | 0 | 0 | **1** | 0 | 2 | 50.00 % |
| Snow/ice | 0 | 0 | 0 | 0 | 0 | 0 | **0** | 0 | - |
| Total number | 91 | 276 | 117 | 24 | 14 | 3 | 0 | **525** | |
| PA | 69.23 % | 88.04 % | 77.78 % | 79.17 % | 100.00 % | 33.33 % | - | | **82.10 %** |



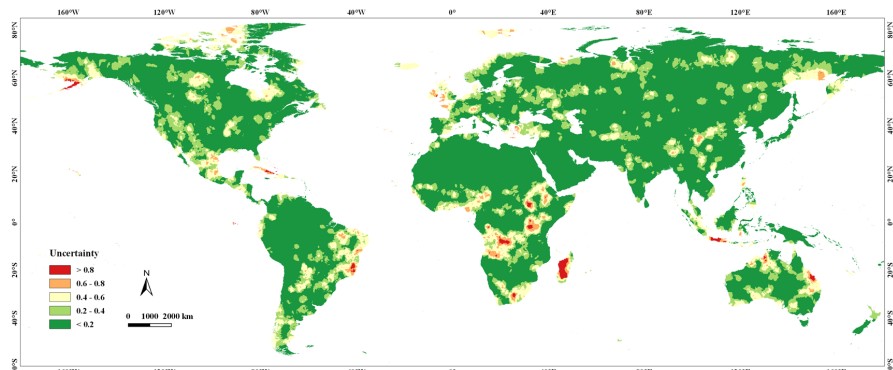

**Figure 7: The geographical distribution of uncertainty for GLASS-GLC (annual dynamics of global land cover) CDRs (Climate Data Records) in 2015, where regions in red represent higher uncertainty levels while those in green represent lower uncertainty levels.**



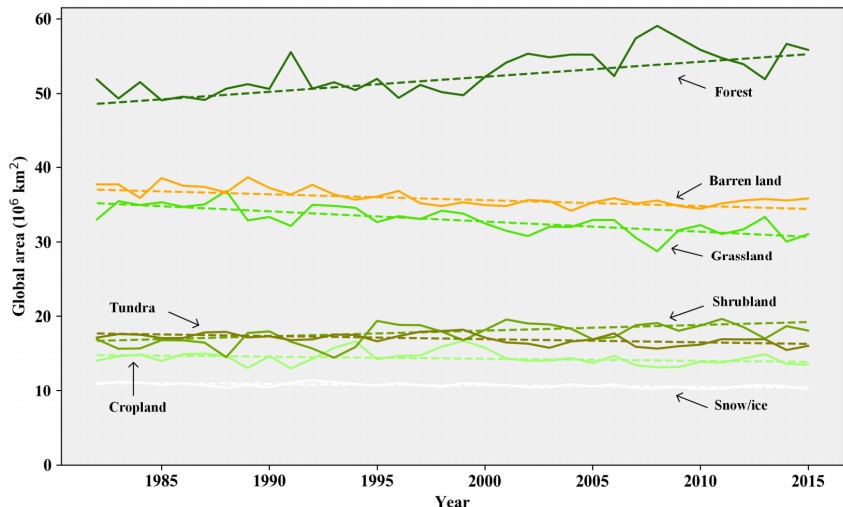

Figure 8: Area curves of global annual land cover change from 1982 to 2015.



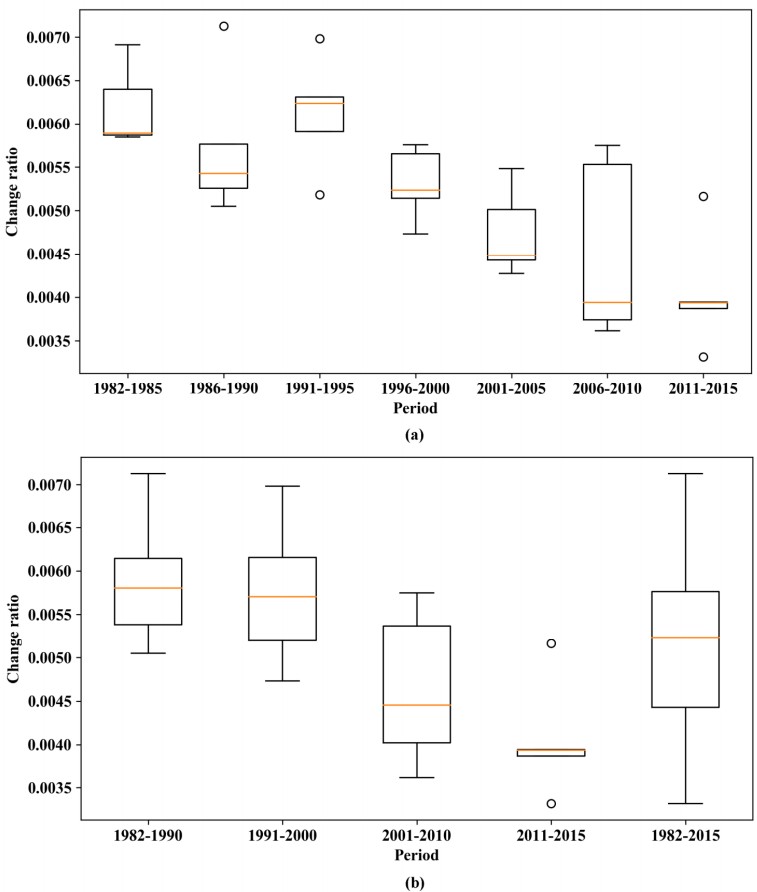


**Figure 9: Comparison and distribution of ratios of annual global land cover change (LCC) to the global total terrestrial land cover**

**(LC) area by different time periods and time intervals (a) 5-year interval, (b) 10-year interval.**





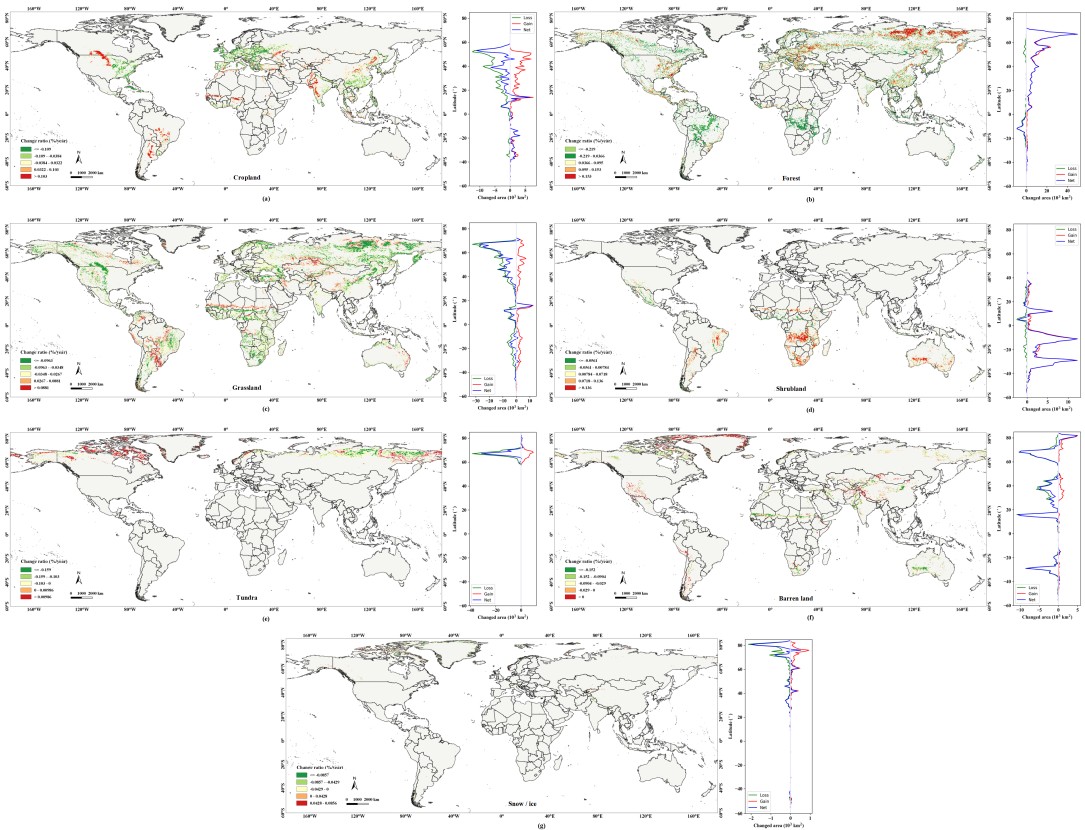

**Figure 10: The geographical distribution of global regions with significant land cover change during 1982-2015, and the summarized**

**results along latitudinal gradients for each class, (a) cropland, (b) forest, (c) grassland, (d) shrubland, (e) tundra, (f) barren land and**

**(g) snow/ice.**



**Table 5: Statistical results of change analysis for cropland (on the scale of continents). Annual change slope and its 95 % confidence**

**interval are given by Thei-sen estimator, p-value and trend information from a Mann-Kendall test. Gain and Loss areas are**

**summarized results relating to the whole time series.**

| Continent | Slope ($10^3$ km²/year) | Lower ($10^3$ km²/year) | Upper ($10^3$ km²/year) | p | Trend | Gain ($10^3$ km²) | Loss ($10^3$ km²) |
|---|---|---|---|---|---|---|---|
| Africa | 5.3 | 1.5 | 10.0 | 0.0099 | increasing | 23 | -6 |
| Asia | -1.7 | -9.2 | 7.1 | 0.6999 | no trend | 67 | -70 |
| Europe | -30.4 | -43.6 | -17.9 | 0.0005 | decreasing | 12 | -99 |
| North America | -4.9 | -10.9 | 2.8 | 0.1635 | no trend | 37 | -54 |
| South America | 9.1 | 2.1 | 19.3 | 0.0108 | increasing | 35 | -4 |
| Oceania | -0.5 | -1.8 | 0.6 | 0.3580 | no trend | 1 | -1 |
| Global | -27.5 | -54.7 | 3.1 | 0.0968 | no trend | 175 | -238 |



**Table 6: Statistical results of change analysis for forest (on the scale of continents) . Annual change slope and its 95 % confidence**

**interval are given by Thei-sen estimator, p-value and trend information from a Mann-Kendall test. Gain and Loss areas are**

**summarized results relating to the whole time series.**

| Continent | Slope ($10^3$ km²/year) | Lower ($10^3$ km²/year) | Upper ($10^3$ km²/year) | p | Trend | Gain ($10^3$ km²) | Loss ($10^3$ km²) |
|---|---|---|---|---|---|---|---|
| Africa | -8.4 | -18.6 | 2.6 | 0.1463 | no trend | 15 | -29 |
| Asia | 128.6 | 86.8 | 165.0 | 0.0000 | increasing | 365 | -12 |
| Europe | 53.1 | 34.9 | 67.4 | 0.0000 | increasing | 131 | -1 |
| North America | 45.1 | 24.7 | 65.0 | 0.0000 | increasing | 132 | -16 |
| South America | -10.8 | -19.6 | -1.4 | 0.0242 | decreasing | 23 | -49 |
| Oceania | 1.4 | -0.1 | 2.6 | 0.0802 | no trend | 6 | -1 |
| Global | 201.3 | 120.9 | 278.1 | 0.0000 | increasing | 680 | -109 |



**Table 7: Statistical results of change analysis for grassland (on the scale of continents) . Annual change slope and its 95 % confidence**

**interval are given by Thei-sen estimator, p-value and trend information from a Mann-Kendall test. Gain and Loss areas are**

**summarized results relating to the whole time series.**

| Continent | Slope ($10^3$ km²/year) | Lower ($10^3$ km²/year) | Upper ($10^3$ km²/year) | p | Trend | Gain ($10^3$ km²) | Loss ($10^3$ km²) |
|---|---|---|---|---|---|---|---|
| Africa | -18.9 | -36.4 | 3.0 | 0.0855 | no trend | 50 | -108 |
| Asia | -52.7 | -67.1 | -38.1 | 0.0000 | decreasing | 85 | -315 |
| Europe | -11.8 | -21.7 | -2.0 | 0.0207 | decreasing | 6 | -59 |
| North America | -39.6 | -48.4 | -26.9 | 0.0000 | decreasing | 25 | -114 |
| South America | -16.1 | -29.0 | -4.7 | 0.0070 | decreasing | 68 | -54 |
| Oceania | -4.6 | -9.5 | 0.7 | 0.1029 | no trend | 9 | -11 |
| Global | -136.6 | -172.9 | -86.4 | 0.0000 | decreasing | 246 | -663 |



**Table 8: Statistical results of change analysis for shrubland (on the scale of continents) . Annual change slope and its 95 % confidence**

**interval are given by Thei-sen estimator, p-value and trend information from a Mann-Kendall test. Gain and Loss areas are**

**summarized results relating to the whole time series.**

| Continent | Slope ($10^3$ km²/year) | Lower ($10^3$ km²/year) | Upper ($10^3$ km²/year) | p | Trend | Gain ($10^3$ km²) | Loss ($10^3$ km²) |
|---|---|---|---|---|---|---|---|
| Africa | 47.4 | 16.1 | 74.8 | 0.0030 | increasing | 120 | -11 |
| Asia | -0.2 | -1.4 | 1.0 | 0.8125 | no trend | 1 | -1 |
| Europe | 0.0 | 0.0 | 0.0 | 0.7561 | no trend | 0 | 0 |
| North America | 0.5 | -3.0 | 5.0 | 0.8356 | no trend | 8 | -7 |
| South America | 17.8 | -0.5 | 34.7 | 0.0618 | no trend | 38 | -6 |
| Oceania | 19.9 | 3.9 | 36.2 | 0.0150 | increasing | 38 | -2 |
| Global | 75.6 | 26.1 | 125.3 | 0.0017 | increasing | 207 | -28 |



**Table 9: Statistical results of change analysis for tundra (on the scale of continents) . Annual change slope and its 95 % confidence**

**interval are given by Thei-sen estimator, p-value and trend information from a Mann-Kendall test. Gain and Loss areas are**

**summarized results relating to the whole time series.**

| Continent | Slope ($10^3$ km²/year) | Lower ($10^3$ km²/year) | Upper ($10^3$ km²/year) | p | Trend | Gain ($10^3$ km²) | Loss ($10^3$ km²) |
|---|---|---|---|---|---|---|---|
| Africa | 0.0 | 0.0 | 0.0 | 1.0000 | no trend | 0 | 0 |
| Asia | -46.7 | -66.6 | -25.3 | 0.0002 | decreasing | 24 | -132 |
| Europe | -4.1 | -6.8 | -2.0 | 0.0015 | decreasing | 3 | -12 |
| North America | 11.4 | 0.6 | 21.5 | 0.0408 | increasing | 42 | -22 |
| South America | 0.0 | 0.0 | 0.0 | 1.0000 | no trend | 0 | 0 |
| Oceania | 0.0 | 0.0 | 0.0 | 1.0000 | no trend | 0 | 0 |
| Global | -42.0 | -63.7 | -20.9 | 0.0019 | decreasing | 71 | -167 |



**Table 10: Statistical results of change analysis for barren land (on the scale of continents) . Annual change slope and its 95 %**

**confidence interval are given by Thei-sen estimator, p-value and trend information from a Mann-Kendall test. Gain and Loss areas**

**are summarized results relating to the whole time series.**

| Continent | Slope ($10^3$ km²/year) | Lower ($10^3$ km²/year) | Upper ($10^3$ km²/year) | p | Trend | Gain ($10^3$ km²) | Loss ($10^3$ km²) |
|---|---|---|---|---|---|---|---|
| Africa | -26.1 | -37.4 | -17.7 | 0.0000 | decreasing | 2 | -43 |
| Asia | -28.3 | -40.6 | -18.1 | 0.0000 | decreasing | 12 | -82 |
| Europe | -2.8 | -3.5 | -1.8 | 0.0000 | decreasing | 0 | -6 |
| North America | -8.8 | -21.3 | -1.0 | 0.0353 | decreasing | 26 | -49 |
| South America | 1.6 | -2.3 | 5.3 | 0.3737 | no trend | 4 | -5 |
| Oceania | -16.8 | -32.2 | 4.0 | 0.1161 | no trend | 0 | -25 |
| Global | -78.5 | -116.4 | -48.8 | 0.0001 | decreasing | 48 | -213 |





**Table 11: Statistical results of change analysis for snow/ice (on the scale of continents) . Annual change slope and its 95 % confidence**

**interval are given by Thei-sen estimator, p-value and trend information from a Mann-Kendall test. Gain and Loss areas are**

**summarized results relating to the whole time series.**

| Continent | Slope ($10^3$ km²/year) | Lower ($10^3$ km²/year) | Upper ($10^3$ km²/year) | p | Trend | Gain ($10^3$ km²) | Loss ($10^3$ km²) |
|---|---|---|---|---|---|---|---|
| Africa | 0.0 | 0.0 | 0.0 | 0.1342 | no trend | 0 | 0 |
| Asia | -2.4 | -4.6 | -0.4 | 0.0117 | decreasing | 2 | -2 |
| Europe | -0.8 | -1.2 | -0.2 | 0.0091 | decreasing | 1 | -1 |
| North America | -12.6 | -20.6 | -6.3 | 0.0015 | decreasing | 4 | -11 |
| South America | -0.2 | -0.3 | -0.2 | 0.0000 | decreasing | 0 | 0 |
| Oceania | 0.0 | -0.1 | 0.0 | 0.0856 | no trend | 0 | 0 |
| Global | -19.2 | -27.6 | -9.1 | 0.0003 | decreasing | 8 | -16 |



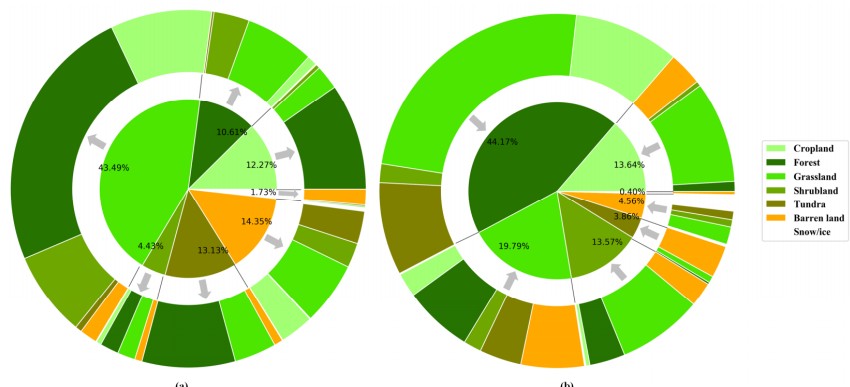

**Figure 11: Land cover conversions with significant land cover change (LCC) between 1982 and 2015, The inner pie in (a) shows the**

**percentages of different gross gain for each land cover, and the outer ring indicates which land cover the gross gain came from. The**

**inner pie in (b) shows the percentage of different gross loss for each land cover, and the outer ring in indicates which land cover the**

**gross loss went to.**



**Table 12: Area ratio (%) of land cover conversions from 1982 to 2015, where the red color denotes a higher ratio, and the blue color**

**represents a lower ratio.**

| Class | | 2015 | | | | | | |
|---|---|---|---|---|---|---|---|---|
| | | Cropland | Forest | Grassland | Shrubland | Tundra | Barren land | Snow/ice |
| 1982 | Cropland | - | 9.6 | 2.22 | 0.37 | 0 | 0.09 | 0 |
| | Forest | 0.9 | - | 6.26 | 3.24 | 0.19 | 0.01 | 0.01 |
| | Grassland | 9.22 | 24.27 | - | 7.73 | 0.6 | 1.6 | 0.06 |
| | Shrubland | 0.45 | 1.7 | 1.62 | - | 0 | 0.66 | 0 |
| | Tundra | 0 | 8.48 | 3.82 | 0 | - | 0.79 | 0.04 |
| | Barren land | 3.07 | 0.07 | 5.75 | 2.23 | 2.93 | - | 0.29 |
| | Snow/ice | 0 | 0.05 | 0.13 | 0 | 0.13 | 1.43 | - |

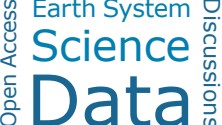



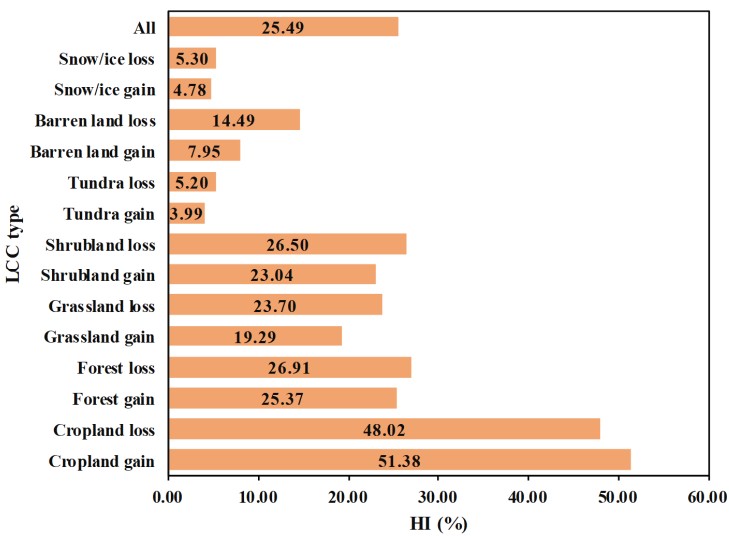

Figure 12: The mean human impact (HI) of regions with significant land cover change (on the scale of LCC).



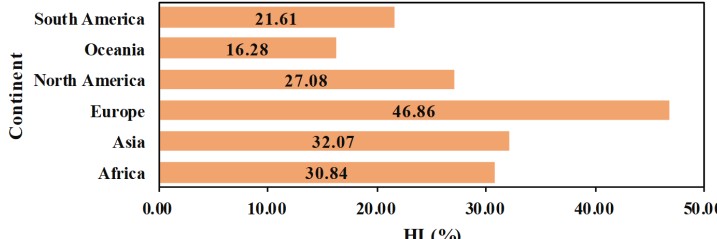

**Figure 13: The mean human impact (HI) of regions with significant land cover change (on the scale of continents).**





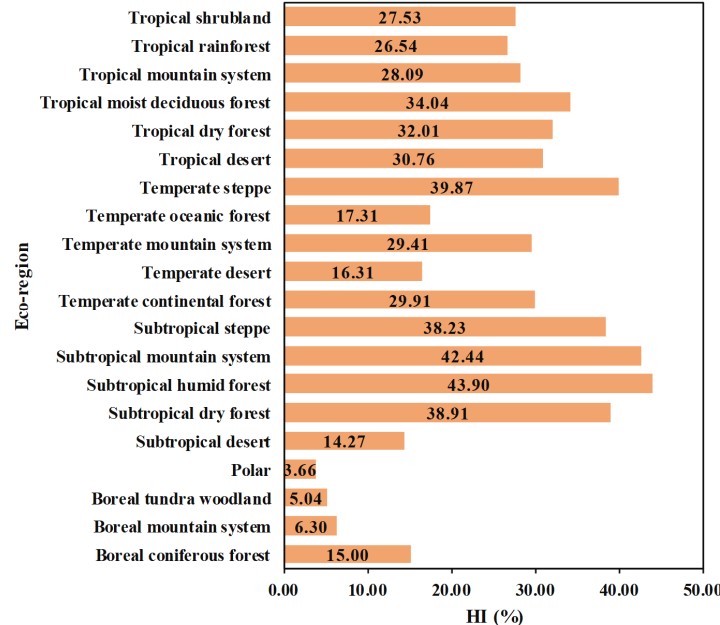


**Figure 14: The mean human impact (HI) of regions with significant land cover change (on the scale of eco-regions).**



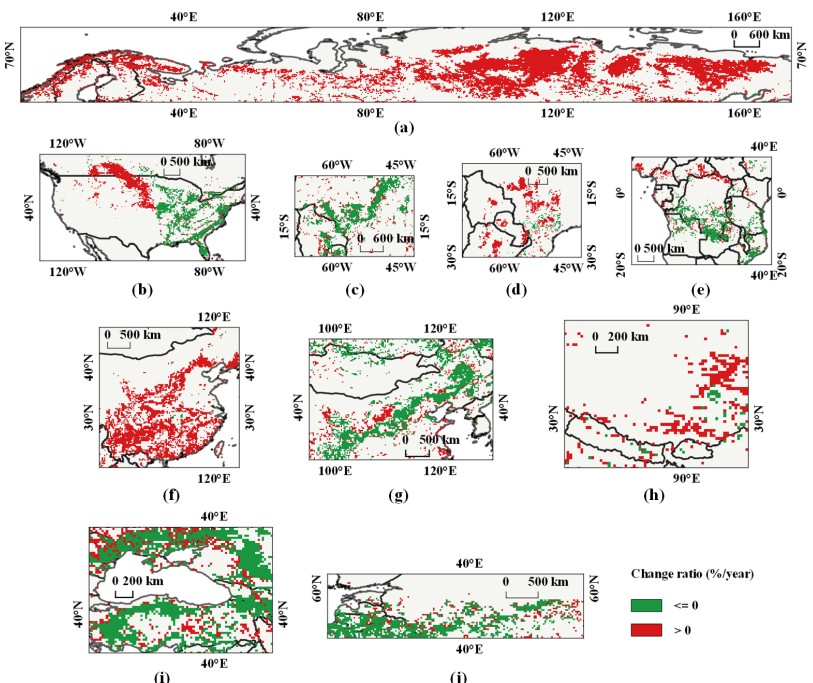

**Figure 15: Visualization of local hotspots of land cover change, (a) north Eurasia, forest, (b) Great Plains of Central North America,**

**cropland, (c) South America, forest, (d) South America, cropland, (e) Africa, forest, (f) China, forest, (g) Mongolia and Inner**

**Mongolia of China, grassland, (h) Qinghai-Tibet Plateau, grassland, (i) central Asia, grassland, (j) the former Soviet Union in**

**Eastern Europe, cropland.**



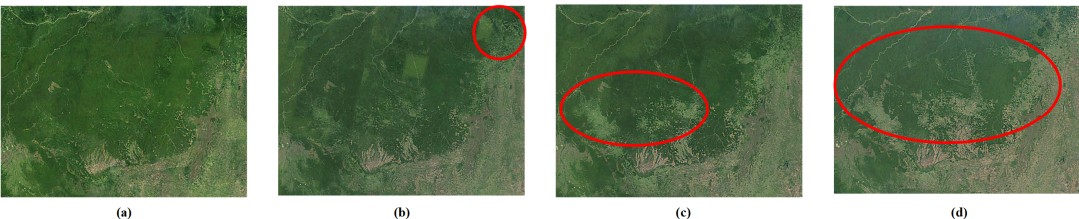

**Figure 16: Example visualization for the cropland expansion and deforestion phenomenon in the southeastern part of the Amazon rainforest from Google Earth images in (a) 1984, (b) 1994, (c) 2000, (d) 2015, where the phenomenon is significant in area within red circles.**