# Peer review of "Annual Dynamics of Global Land Cover and its Long-term Changes from 1982 to 2015"

_Earth System Science Data, 2019_

## Referee Comment (RC1) · Anonymous Referee #1 · 30 Apr 2019

The authors used GLASS CDRs data and Google Earth Engine platform and produced the long-term continuous land cover dataset from 1982 to 2015. This is a very valuable dataset for further applications in the analyses of energy and carbon dynamics and the global land surface modelling. However, I have some concerns about the data processing in the classification, accuracy assessment and the interpretation of the land cover change results. I think these problems must be solved / addressed before publication.

1. Differences between forest and tree cover

The authors used vegetation cover fraction (VCF) data from Song et al. 2018. However, in their paper, they specified "tree cover" increase. This is not equal to forest increase. Usually, the forest is defined by canopy closure (e.g. tree cover fraction >10% in FAO, >25% in Hansen et al. 2013), tree height and minimum area. The authors showed that a lot of forest increase occurred in Siberia (Fig. 10) and was from grassland (Fig. 11). This could be artificial considering the coarse resolution (5 km) and poor ability of LC mapping for mosaic pixels (see below) in the methods used in this study. For example, there is 5 ha forest with tree cover fraction of 35%, and the tree cover fraction increased to 45% because of better growth (e.g. longer growing season, CO2) in the same 5 ha forest. In this case, we cannot say the forest area increased 5 ha x 10% = 0.5 ha because it is the same 5 ha forest but with denser tree cover. Therefore, I doubt that there is confusion of these concepts in this manuscript and maybe in the classification system. The authors briefly mentioned this issue on L450-454, but this really needs to be clarified, assessed and solved.

2. The majority LC in a 0.05 deg pixel

The majority method in a coarse resolution (5 km) may work for some pure pixels but is expected to work poorly for the mosaic pixels with high heterogeneity or similar fraction of different vegetation types. For example, in a 5 km pixel with 43% tree cover, 44% grass and 13% others in the first year, it became 45% tree cover, 44% grass and 11% of others in the second year simply because of the good climate. If I understood correctly, this pixel would be classified as grassland in the first year and forest in the second year, and thus there is a 25 km2 land cover change from grassland to forest. This may also partly explain the strong forest increase in Siberia, high variations in the temporal LC dynamics in Fig. 8 and the high uncertainties in the intensive LCC regions (e.g. savanna in Africa).

3. Accuracy of change detection

The authors only assessed the accuracy for year 2015, not mentioning that the uncertainty of FROM-GLC_v2 was not propagated. First, the same product was used for training the classification system and for the accuracy assessment. Although the samples in the same product may be not overlapped, we cannot exclude the coherence since both are from FROM-GLC_v2. So, some independent evaluation dataset would be helpful. Second, an important feature of this continuous LC maps is the temporal dynamics. So, the change detection needs to be further validated / evaluated in addition to the one-year classification accuracy assessment. This part is currently lacking in this work.

4. Comparison with other datasets

A suggestion for the evaluation may be to compare the total area, spatial and temporal changes with other datasets e.g. ESA-CCI 300 m, Hansen forest, FAO and some cropland datasets. This would help to verify the mapping results in this study and to understand their differences. It would also

help to define the possible applications of this dataset (e.g. whether it can be used for carbon accounting, land modeling).

5. Superficial and fragmented interpretations of reasons for LCC

The authors made a lot of figures and tables to show the spatial and temporal changes and also reasons for such changes. These sections are not well organized and lack some internal logics. What I learned is only some fragmented information. The reasons for the LCC are not very solid (see my detailed comments below). Just taking one example, the author mentioned several times of "greening" and its effects on LCC. However, greening is very far away from LCC. It may only be caused by more leaves and extended growing season. We don't know whether this increased productivity was converted to carbon stock or leaded to a LC transition from grass to forest. The increased carbon uptake by greening may just release back to the atmosphere through the enhanced respiration due to increased temperature. So, I would suggest being cautious when interpreting the reasons for the LCC. In fact, I don't think these sections are necessary for this manuscript. Adding comparisons with other datasets and discussing the differences between various data and the reasons (e.g. data sources, classification methods) would be enough for a nice data paper. The reasons for LCC can be separated to another paper after adding more analyses. Putting it here only attenuate the main objective of this manuscript.

6. Writing

Language needs further improvements. A lot of sentences are difficult to understand, and some sentences are broken in the context. Please polish the language during revision.

Specific comments:

L19: report how many classes

L20: 85% accuracy based on what?

L22: how can you separate afforestation and forest expansion?

L23: land degradation? did you mean grassland loss? if it is degradation, it may still be grassland.

L25: greening is not directly related to LCC. very complex processes behind.

L37: What is "surface attributes"?

L44: too strong statement

L70: "which will…" useless half sentence

L75: not clear "more prone to consistency and data volume", rephrase

L87: "Because of…" duplicate

L91: analyses

L136: explain if you have level 2 class and how they were derived

L142: "2013-2015" is it a one-year map or three maps each for a year?

L145: "with a limited …" not clear what it is.

L147: "class distribution" do you mean "percentage of each class"?

L158: what is end-number? end of what?

L162: Is the smaller fluctuation the truth? Something you expected?

L172: How is the performance of you trained random forest classifier? OOB R2 or independent evaluation dataset?

L174: what are the other parameters and the default values?

L179: "the mode of…" not clear

L186: How about the heterogeneous pixels? Not assessed at all?

L190: "class distribution"

L206: It is OK to fit a linear trend, but you cannot say to remove … because it may be caused by the actual LCC

L212: why is summed?

L213: what is "annual change in slope of area ration"?

L219: why only statistically significant change was included? It is still area change even the trend is not significant. The way you process data exaggerate the changes.

L223-224: Again, why only change mask?

L225: "direct" duplicate

L242: Need to explain UA and PA for non-remote-sensing readers; explain what the column and row names refer to.

L245: "Grassland is …" , from Table 3, they are shrubland and forest

L248: samples

L255: these are regions with intensive LCC

L259: "variation curves" temporal changes

L260: Why so strong forest increase from 2006-2008? is it real?

L262: what about cropland? why so high variations, especially in 1994, 1999?

L263: Fig. 9, explain the meaning of your boxplot, mean, median, IQR, 90%, max, min? Why use the ratio, instead of total change area which is more straightforward?

L263: "different time periods" the gross change each year or on the difference between the first and the last year in each period?

L267: It's interesting to see a very likely decreasing trend of total LCC area.

L270: Fig. 10: The text and subplots in the figure is too small to read. I would suggest to only show the main LC types and put the others to SI

L271: why only significant LCC? is it really necessary? why not just sum all?

L287: Table 5-10: too detailed, may put into SI and merge these results in a plot with different subplots

L299: In In

L304-308: see my comments on greening above

L309: need to note the high uncertainty from Fig. 7

L310-311: greening again

L313: Look at Fig. 10 and 11, significant grassland changed to forest in your dataset in the high latitudes

L316-317: why barren land decrease implies the desertification effects?

L321: what is a coupling effect? non-relevant sentence

L324 and all below: referring a or b when you report something. Why no explanations on the transitions from grassland to forest, which is the most obvious pattern in your figure

L338: too strong statement. surface greening is not something that you can directly interpreted from LCC.

L345: "natural vegetation" managed forest or pasture are not natural vegetation

L346: how about reforestation?

L355: shy subtropical mountain system is also high?

L363: Fig. 15 is very misleading with only >0 and <0. Why not give gradient of change?

L365 and below: again, give the subplot title when you describing the results.

L381: Do you have evidence that global warming will increase vegetation in tropics?!

L383-384: Oil palm plantations are forest or crop in your classification system? I am not sure whether you can distinguish them!

L399: Is that partly why you detected forest increase at the expense of grassland?

L421-423: yes, this is the main defect of this product.

L435: This is definitely something that has to be done in this work.

L441: what about the heterogeneous pixels?

L460: NDVI and LAI increase not equal to forest increase

L455-463: not helping but expose the weakness of the product

L464-465: This contradicts that you said forest loss in SE Asia is due to oil palm plantations

L466-467: need more explanations to justify the reasons for doing this.

---

## Referee Comment (RC2) · Anonymous Referee #2 · 23 Nov 2019

General comment: By fusing multiple existing geo-spatial datasets, the main work of this manuscript is to generate an annual dynamic product (spatial resolution: 0.5°) addressing seven kinds of land-covers (i.e., cropland, forest, grassland, shrub-land, tundra, barren land and snow/ice) from 1982 to 2015. With eye on the current existing datasets (i.e., from the perspective of classification system, period of time, and spatial/temporal resolution) the contribution is quite limited. In view of the rationality of technique and accuracy assessment, current version calls for serious revision before publication. In view of the analysis conducted on the dynamic map, rare novel findings can be captured. Specific comments: There are several global datasets with more rigorous production process have existed. 1) The 1992-2018 annual 300m global land-cover data (https://www.esa-landcover-cci.org/?q=node/197 ) with more detailed

classification scheme have been released. Since the proposed product has no accuracy assessment on the annual maps from 1982-1991, it cannot be argued that the proposed work have longer period of time. 2) The annual VCF products from 1982-2016 have the same spatial resolution, and very similar classification scheme with the proposed work (from 1982-2015). Although the VCF product is missing in 1994 and 2000, the proposed work just directly use the data-source around the adjacent year, which cannot be viewed as a noticeable contribution. Meanwhile, since the proposed work also introduce VCF in the supervised classification, the analysis on the dynamic map is somewhat similar to this existed study (Song et al 2018a) , but more superficial.

Technical corrections 1. It is ridiculous to produce training and test set from a same product and in a same manner. In addition, it is unacceptable to conclude the applicability of the long-time period product by assessing the accuracy on only the 2015 land-cover mapping result. 2. How to project the 30m FROM-GLC_v2 to mapping scale? How to deal with the mixed sample? 3. There is no sample accuracy assessment on the produced training sample set. Please note that the accuracy of the FROM-GLC_v2 is not high enough to work as training sample. 4. When mapping the land-covers decades year ago, the suitability of the samples collected (mainly from 2013-2015) should be evaluated.

---

## Author Comment (AC1) · 18 Jan 2020

The authors used GLASS CDRs data and Google Earth Engine platform and produced the long-term continuous land cover dataset from 1982 to 2015. This is a very valuable dataset for further applications in the analyses of energy and carbon dynamics and the global land surface modelling. However, I have some concerns about the data processing in the classification, accuracy assessment and the interpretation of the land cover change results. I think these problems must be solved / addressed before publication.

We thank the reviewer for the comments and thoughtful review. Please find our detailed response along with the suggested changes to our manuscript below.

1. Differences between forest and tree cover
The authors used vegetation cover fraction (VCF) data from Song et al. 2018. However, in their paper, they specified "tree cover" increase. This is not equal to forest increase. Usually, the forest is defined by canopy closure (e.g. tree cover fraction >10% in FAO, >25% in Hansen et al. 2013), tree height and minimum area. The authors showed that a lot of forest increase occurred in Siberia (Fig. 10) and was from grassland (Fig. 11). This could be artificial considering the coarse resolution (5 km) and poor ability of land cover mapping for mosaic pixels (see below) in the methods used in this study. For example, there is 5 ha forest with tree cover fraction of 35%, and the tree cover fraction increased to 45% because of better growth (e.g. longer growing season, $CO_2$) in the same 5 ha forest. In this case, we cannot say the forest area increased 5 ha x 10% = 0.5 ha because it is the same 5 ha forest but with denser tree cover. Therefore, I doubt that there is confusion of these concepts in this manuscript and maybe in the classification system. The authors briefly mentioned this issue on L450-454, but this really needs to be clarified, assessed and solved.

**Response 1**:

Thank you for your advice. It should be pointed out that our classification target is land cover class, not vegetation cover percentage information. Our land cover products belong to the hard classification and give each mapping unit a single land cover class. VCF is only used as features that assist in the land cover classification, which is introduced as prior probability, and only one of many factors that affect the final classification result. Although based on VCF information, our results are not the same.

The classification system we used is from FROM-GLC_v2 (Li et al., 2017). Considering the quality of the data and the separability of classes, our products include 7 land cover classes, cropland, forest, grassland, shrubland, tundra, barren land, and snow and ice (Table 1). Among them, the forest is also defined and distinguished by canopy closure. The forest is defined under the condition that tree cover≥10% and height> 5m. We have updated the description of the classification system in our manuscript.

Table 1: Classification system, with 7 Level 1 classes and 21 Level 2 classes.

| Level 1 class | Level 2 class | Description |
|---|---|---|
| Cropland | Rice paddy
Greenhouse
Other farmland
Orchard
Bare farmland | |
| Forest | Broadleaf, leaf-on
Broadleaf, leaf-off
Needle-leaf, leaf-on
Needle-leaf, leaf-off
Mixed leaf type, leaf-on
Mixed leaf type, leaf-off | Tree cover≥10%;
Height>5m;
For mixed leaf, neither coniferous nor broadleaf types exceed 60% |
| Grassland | Pasture, leaf-on
Natural grassland, leaf-on
Grassland, leaf-off | Canopy cover≥20% |
| Shrubland | Shrub cover, leaf-on
Shrub cover, leaf-off | Canopy cover≥20%;
Height<5m |
| Tundra | Shrub and brush tundra
Herbaceous tundra | |
| Barren land | Barren land | Vegetation cover<10% |
| Snow/Ice | Snow
Ice | |

We agree with you that forest increase may exist under the condition that you described. This is an inevitable problem in hard classification. What we call forest increase is the change of land cover class in our classification results under our 5km coarse resolution classification system. Limited to a spatial resolution of 5km, there are many mixed mapping units. For these mixed units, the estimation of hard classification will cause a large deviation. This is a common problem in hard classification. Similar problems also exist in land cover data prediction with higher spatial resolution. At coarse resolution, accurate estimates may be better with cover percentage data.

**Change in manuscript:**

We have updated the description to our used classification system in Table 1.

2. The majority land cover in a 0.05 deg pixel
The majority method in a coarse resolution (5 km) may work for some pure pixels but is expected to work poorly for the mosaic pixels with high heterogeneity or similar fraction of different vegetation types. For example, in a 5 km pixel with 43% tree cover, 44% grass and 13% others in the first year, it became 45% tree cover, 44% grass and 11% of others in the second year simply because of the good climate. If I understood correctly, this pixel would be classified as grassland in

the first year and forest in the second year, and thus there is a 25 km2 land cover change from grassland to forest. This may also partly explain the strong forest increase in Siberia, high variations in the temporal land cover dynamics in Fig. 8 and the high uncertainties in the intensive LCC regions (e.g. savanna in Africa).

**Response 2**:

Thanks for your comment. For mosaic pixels, especially mosaic pixels of vegetation, hard classification does have such disadvantages. The classification system used in the MODIS-based land cover product has included some mosaic classes, such as the Forest / Cropland Mosaics, Natural Herbaceous / Croplands Mosaics and Herbaceous Croplands defined in the FAO-Land Cover Classification System land use (LCCS2) system, which also reflects the difficulty and disadvantage of hard classification in coarse resolution to a certain extent. However, for these mosaic classes in the MODIS-based land cover product, hard classification is still used. Although at individual pixel level this is unavoidable when land cover data are aggregated over large areas the extreme cases as raised by the reviewer would usually be averaged out.

Despite of the disadvantage, the way that hard classification presents information is more direct. In many applications, researchers prefer to use the results of hard classification.

Besides, the scheme we used to aggregate and extract coarse-resolution samples from fine-resolution data is one of the common used schemes (DeFries et al., 1998;Wang et al., 2016). Under the framework of hard classification, there does not seem to be a better solution.

As for the LCC area reflected in the product, there are some places, as you said, that may be affected by the hard classification method. However, there are also many areas where the LCC is correctly reflected, such as the forest area of the Amazon region cut back. It should be pointed out that the LCC information in our results has uncertainty, especially the regions with high variability in LCC.

**Change in manuscript:**

We have added a reminder to data users about the uncertainty of our products.

3. Accuracy of change detection
The authors only assessed the accuracy for year 2015, not mentioning that the uncertainty of FROM-GLC_v2 was not propagated. First, the same product was used for training the classification system and for the accuracy assessment. Although the samples in the same product may be not overlapped, we cannot exclude the coherence since both are from FROM-GLC_v2. So, some independent evaluation dataset would be helpful. Second, an important feature of this continuous land cover maps is the temporal dynamics. So, the change detection needs to be further validated / evaluated in addition to the one-year classification accuracy assessment. This part is currently lacking in this work.

**Response 3**:

Thank you for your useful comment. In this revision, we collected a new independent test sample and performed the accuracy assessment. To prove the impact of change detection, we further compared the accuracies with and without change detection.

Specifically, we collected 2431 randomly distributed 5km sample points in different years around the world. According to the majority principle, we manually interpreted the land cover class of each sample as an independent test sample. Besides, to verify the accuracy of the change detection method, we also compared the classification accuracy before and after the change detection. The temporal distribution of the newly collected test samples is shown in Fig. 1, and the geographical distribution is shown in Fig. 2.

[Figure]

Figure 1: The temporal distribution of the newly collected test sample.

[Figure]

Figure 2: The geographical distribution of random test sample.

The new assessment result is shown in Table 3 and Table 4. It shows that OA of GLASS-GLC without change detection is 81.28%, and OA with change detection is 82.81%. This reflects the reliability of GLASS-GLC since the test samples are randomly distributed along the spatial and temporal dimensions, and also confirm the significance and effectiveness of the change detection method.

Table 3: Classification accuracy of GLASS-GLC without change detection under 2431 independent test samples. (Overall accuracy = 81.28 %, UA = User's Accuracy and PA = Producer's Accuracy)

| Class | Cropland | Forest | Grassland | Shrubland | Tundra | Barren land | Snow/ice | Total number | UA |
|---|---|---|---|---|---|---|---|---|---|
| Cropland | 257 | 21 | 34 | 15 | 0 | 31 | 0 | 358 | 71.79% |
| Forest | 35 | 620 | 45 | 27 | 22 | 1 | 1 | 751 | 82.56% |
| Grassland | 17 | 26 | 248 | 12 | 3 | 19 | 4 | 329 | 75.38% |
| Shrubland | 7 | 6 | 10 | 154 | 9 | 12 | 0 | 198 | 77.78% |
| Tundra | 0 | 9 | 11 | 12 | 250 | 3 | 0 | 285 | 87.72% |
| Barren land | 4 | 1 | 13 | 14 | 5 | 355 | 6 | 398 | 89.20% |
| Snow/ice | 0 | 4 | 3 | 0 | 0 | 13 | 92 | 112 | 82.14% |
| Total number | 320 | 687 | 364 | 234 | 289 | 434 | 103 | 2431 | |
| PA | 80.31% | 90.25% | 68.13% | 65.81% | 86.51% | 81.80% | 89.32% | | 81.28% |

Table 4: Classification accuracy of GLASS-GLC with change detection under 2431 independent test samples. (Overall accuracy =82.81 %, UA = User's Accuracy and PA = Producer's Accuracy)

| Class | Cropland | Forest | Grassland | Shrubland | Tundra | Barren land | Snow/ice | Total number | UA |
|---|---|---|---|---|---|---|---|---|---|
| Cropland | 262 | 19 | 32 | 20 | 0 | 25 | 0 | 358 | 73.18% |
| Forest | 33 | 637 | 29 | 28 | 24 | 0 | 0 | 751 | 84.82% |
| Grassland | 24 | 24 | 254 | 6 | 13 | 8 | 0 | 329 | 77.20% |
| Shrubland | 12 | 3 | 11 | 159 | 6 | 7 | 0 | 198 | 80.30% |
| Tundra | 0 | 12 | 9 | 4 | 250 | 10 | 0 | 285 | 87.72% |
| Barren land | 5 | 1 | 17 | 8 | 7 | 357 | 3 | 398 | 89.70% |
| Snow/ice | 0 | 5 | 6 | 0 | 0 | 7 | 94 | 112 | 83.93% |
| Total number | 336 | 701 | 358 | 225 | 300 | 414 | 97 | 2431 | |
| PA | 77.98% | 90.87% | 70.95% | 70.67% | 83.33% | 86.23% | 96.91% | | 82.81% |

**Change in manuscript:**

We have added the new accuracy assessment result in the manuscript.

4. Comparison with other datasets

A suggestion for the evaluation may be to compare the total area, spatial and temporal changes with other datasets e.g. ESA-CCI 300 m, Hansen forest, FAO and some cropland datasets. This would help to verify the mapping results in this study and to understand their differences. It would also help to define the possible applications of this dataset (e.g. whether it can be used for carbon accounting, land modeling).

**Response 4**:

Thank you for your advice. Comparison with other land cover products is a very good way to reflect product quality and accuracy. For this reason, in addition to the classification accuracy obtained by several evaluation methods, we compared other available land cover products with our products. Although there are some differences in the classification system of different products, it can still reflect the reliability of our products in general.

We inter-compared GLASS-GLC with other available global land cover products with a relatively long time series. Land cover products from MODIS and the ESA-CCI were used. The MODIS-based global land cover products come from Collection 6 (C6) MODIS Land Cover Type (MLCT) products (Sulla-Menashe et al., 2019), and are supervised classification results from 2001 to 2016. Considering the comparability to our classification system, the FAO-Land Cover Classification System land use (LCCS2) layer was used. The corresponding relationships of classes are listed as follows, and the class names we used are the latter: barren - barren land, permanent snow and ice – snow/ice, all kinds of forest – forest, forest/cropland mosaics and natural herbaceous/cropland mosaic – cropland, natural herbaceous and herbaceous cropland – grassland, shrubland - shrubland. The ESA-CCI global land cover products (Bontemps et al., 2013) are 300m resolution yearly products ranging from 1992 to 2015. The products were developed using the GlobCover unsupervised classification chain and merging multiple available Earth observation products based on the GlobCover products of the ESA (Liu et al., 2018). Referring to the class relationships in (Liu et al., 2018), we cross-walked classes including cropland, forest, grassland, shrubland, barren land and snow/ice.

[revised manuscript text omitted]

**Change in manuscript:**

We have included comparison results with other land cover data in the manuscript to help show the reliability and effectiveness of our products.

5. Superficial and fragmented interpretations of reasons for LCC

The authors made a lot of figures and tables to show the spatial and temporal changes and also reasons for such changes. These sections are not well organized and lack some internal logics. What I learned is only some fragmented information. The reasons for the LCC are not very solid (see my detailed comments below). Just taking one example, the author mentioned several times of "greening" and its effects on LCC. However, greening is very far away from LCC. It may only be caused by more leaves and extended growing season. We don't know whether this increased productivity was converted to carbon stock or leaded to a land cover transition from grass to forest. The increased carbon uptake by greening may just release back to the atmosphere through the enhanced respiration due to increased temperature. So, I would suggest being cautious when interpreting the reasons for the LCC. In fact, I don't think these sections are necessary for this manuscript. Adding comparisons with other datasets and discussing the differences between various data and the reasons (e.g. data sources, classification methods) would be enough for a nice data

paper. The reasons for LCC can be separated to another paper after adding more analyses. Putting it here only attenuate the main objective of this manuscript.

**Response 5**:

Thank you for your comment. The interpretations of reasons for LCC are just some examples of our attempts to apply our product for further analysis, not the main focus of this paper. The focus of this article is on the presentation and quality assessment of our produced GLASS-GLC data products. To this end, we have added more content on accuracy assessment and product inter-comparison, to better demonstrate the reliability and uncertainty of our products. As for the reasons for LCC, we will analyze and discuss in more detail in a subsequent paper.

**Change in manuscript:**

We have supplemented the sections of accuracy assessment and data inter-comparison.

6. Writing
Language needs further improvements. A lot of sentences are difficult to understand, and some sentences are broken in the context. Please polish the language during revision.

**Response 6**:

Thanks for your suggestion.

**Change in manuscript:**

We have polished our language with a native English consultant.

Specific comments:
L19: report how many classes

**Response s1**:

The classification system consists of 7 classes, including cropland, forest, grassland, shrubland, tundra, barren land, snow/ice, as shown in Table 1 in the manuscript.

**Change in manuscript:**

We have added the information in the corresponding place as you suggested.

L20: 85% accuracy based on what?

**Response s2:**

It was based on 23459 test samples in 2015. And the overall accuracy of the produced GLASS-GLC CDR in 2015 is 86.51 %. The test samples come from the 30 m resolution FROM-GLC_v2 test sample set (Li et al., 2017).

To give a more effective assessment, we also performed an accuracy assessment using FLUXNET site data and the newly collected independent test samples, and we supplemented this part of the results.

**Change in manuscript:**

We have updated the detail in the corresponding place.

L22: how can you separate afforestation and forest expansion?

**Response s3:**

The data products we produce can only provide information at the observation level. For example, the information we can obtain here is only forest gain. While the specific causes of these LCCs should be analyzed and investigated separately, we cannot distinguish afforestation and natural expansion of forests.

**Change in manuscript:**

We have modified our description according to our study.

L23: land degradation? did you mean grassland loss? if it is degradation, it may still be grassland.

**Response s4:**

Yes, we do mean by grassland loss. At the individual mapping unit level, we cannot detect land degradation.

**Change in manuscript:**

We have changed the word "land degradation" to "grassland loss".

L25: greening is not directly related to LCC. very complex processes behind.

**Response s5:**

Greening is indeed a very complex process. Here, we mainly refer to vegetation gain such as forest gain in our results, which can only be used as evidence from the perspective of remote sensing and mapping. Thanks for pointing it out.

**Change in manuscript:**

We have corrected our expression.

L37: What is "surface attributes"?

**Response s6:**

It refers to the characteristics and properties of the Earth surface. The change of land cover would change the status of the Earth surface.

**Change in manuscript:**

We have changed the word "attributes" to "characteristics".

L44: too strong statement

**Response s7:**

Thank you for your reminding.

**Change in manuscript:**

We have modified our expression.

L70: "which will…" useless half sentence

**Response s8:**

Many thanks.

**Change in manuscript:**

We have rechecked the sentence and deleted it.

L75: not clear "more prone to consistency and data volume", rephrase

**Response s9:**

What we mean here was that Landsat data has a higher spatial resolution, but it also meets some problems including obvious cloud contamination, data inconsistency caused by multiple generations of sensors and relatively larger data volume because of the high resolution.

**Change in manuscript:**

We have changed it to detailed description.

L87: "Because of…" duplicate

**Response s10:**

Thanks for your comment.

**Change in manuscript:**

We have changed the used "Because of" to "Due to".

L91: analyses

**Response s11:**

Thank you for correcting us.

**Change in manuscript:**

We have changed the word.

L136: explain if you have level 2 class and how they were derived

**Response s12:**

There are no Level 2 classes in our results. Considering the resolution and separability of GLASS data, only Level 1 classes are included. The description of Level 2 classes comes from the original design in the FROM-GLC classification system (Gong et al., 2013). It was listed in Table 1 to better show the meaning of each Level 1 class. In the future, we will try to produce land cover products

with a more detailed classification system.

L142: "2013-2015" is it a one-year map or three maps each for a year?

**Response s13:**

It was a one-year map, not three maps. Due to the problem of data quality, the Landsat data in one year usually cannot meet the need for land cover mapping on a global scale. The production of the FROM-GLC_v2 map took advantage of data from 2013 to 2015. And it can roughly be called circa 2015 (Li et al., 2017).

L145: "with a limited …" not clear what it is.

**Response s14:**

When generating random points in ArcGIS with the "create random points" tool, we limited the spatial interval among points greater than 0.1° by setting the parameter "minimum allowed distance" as 0.1.

**Change in manuscript:**

We changed our description.

L147: "class distribution" do you mean "percentage of each class"?

**Response s15:**

Yes. It means the percentage of each class.

**Change in manuscript:**

We have changed the description.

L158: what is end-number? end of what?

**Response s16:**

It is called end-member. Due to the complexity of ground objects and the limited spatial resolution of various sensors, the information contained in a pixel of remote sensing images is the mixture of information of many ground objects, hence resulting in mixed pixels (Zhang et al., 2011). It is

assumed that there are pure land cover types known as basic mixing elements (known as end-member) that cannot be further decomposed in the imaged area, and the process of finding these end-members is referred to endmember extraction (Plaza et al., 2002). Here, to reduce the systematic deviation of AVHRR products, we correct the GLASS data with MODIS products based on end-members (Song et al., 2018).

L162: Is the smaller fluctuation the truth? Something you expected?

**Response s17:**

It is a trade-off. The purpose of data correction was to correct the original remotely sensed data to a higher consistency, especially in the temporal dimension. Remotely sensed data is easy to be affected by many random and systematic factors such as the atmospheric environment and sensor situation. The values it reflects were usually not those of the real and direct surface conditions. What's more, many fake inter-annual variations exist in remotely sensed data (Friedl et al., 2010). This will cause much trouble and disturbance especially in the use of time-series remotely sensed data. Though the variations may be caused by phenological changes, other interfering factors exist, and the trade-off is more beneficial in general.

The correction process carried out belongs to one of the data pre-processing processes in time-series land cover mapping (Gómez et al., 2016), which was to mitigate and deal with these aspects and to produce more consistent data for use.

L172: How is the performance of you trained random forest classifier? OOB R2 or independent evaluation dataset?

**Response s18:**

The OOB accuracy of our random forest classifier reached to 87.12%.

**Change in manuscript:**

We have added this information in the manuscript.

L174: what are the other parameters and the default values?

**Response s19:**

The specific parameters are listed as follows. The number of trees was 200, the out-of-bag mode is on. The number of variables per split was set to 0, as the square root of the number of variables. The minimum size of a terminal node was 1, the fraction of input to bag per tree was 0.5, and the random

seed was 0.

Table 8 Specific parameters of the random forest classifier

| Parameter | Value |
|---|---|
| Number of trees | 200 |
| Number of variables per split | 0 |
| Minimum size of a terminal node | 1 |
| Fraction of input to bag per tree | 0.5 |
| Whether the classifier should run in out-of-bag mode | True |
| Random seed | 0 |

**Change in manuscript:**

We have listed the above parameter values in the manuscript.

L179: "the mode of…" not clear

**Response s20:**

The mode here refers to the class label that has the highest frequency in the segmented period with the calculated breakpoints. To improve the time consistency in the classification results, we use the mode class label to replace all the class labels in the period.

**Change in manuscript:**

We have updated our expression.

L186: How about the heterogeneous pixels? Not assessed at all?

**Response s21:**

Thanks for your question. In the newly collected independent test sample set, we use random points, with no difference between homogeneous and heterogeneous pixels. Therefore, the new assessment results include heterogeneous pixels.

L190: "class distribution"

**Response s22:**

Thanks for your reminding.

**Change in manuscript:**

We have changed the description.

L206: It is OK to fit a linear trend, but you cannot say to remove … because it may be caused by the actual LCC

**Response s23:**

Thanks for your advice. Our purpose was to fit a linear trend for better extraction of the land cover change trend in the long time-series land cover data. The fluctuations in the land cover were generally seen as an abnormal condition caused by climate conditions and phenological changes since the land cover is stable in most areas in the world across years, but they can be caused by the actual land cover change as you said.

**Change in manuscript:**

We have changed the description.

L212: why is summed?

**Response s24:**

Because we wanted to ensure the significance of the land cover change trend. For each pixel in the land cover map of each class, the original 0.05 ° pixel is labeled with 0 or 1 (belonging to the class or not), and such categorical data (not continuous data) cannot be statistically hypothesized. In order to carry out the hypothesis test, some studies used downscaling (Wang et al., 2016). By downscaling, the categorical label data can be summed up as the area ratio of the class (numerical data) in a greater statistical area, thus a statistical hypothesis test can be performed to verify the significance of land cover change.

L213: what is "annual change in slope of area ration"?

**Response s25:**

We are sorry for making a slip in writing. It is in fact "annual change slope of area ratio" estimated from a Theil-Sen estimator. More specifically, it represents the speed of land cover change.

**Change in manuscript:**

We have corrected it.

L219: why only statistically significant change was included? It is still area change even the trend is not significant. The way you process data exaggerate the changes.

**Response s26:**

Yes, there are certain shortcomings in doing so. But relatively speaking, this is a better strategy. Because it usually exists fake inter-annual land cover change in time-series land cover mapping studies (Sulla-Menashe et al., 2019) caused by many kinds of factors as explained in the above. Although there may be some real land cover change, to ensure the significance and reduce the uncertainty we did not include those into the statistics.

L223-224: Again, why only change mask?

**Response s27:**

Because we want to ensure the statistical significance and reduce the uncertainty caused by classification noises to detect more robust long-term land cover change trends.

L225: "direct" duplicate

**Response s28:**

Thank you for your comment.

**Change in manuscript:**

We have deleted the word.

L242: Need to explain UA and PA for non-remote-sensing readers; explain what the column and row names refer to.

**Response s29:**

Thanks for reminding. UA and PA represent user's accuracy and producer's accuracy respectively. They are two metrics reflecting the accuracy of classification. UA = corrected classified sample number / total sample number in the classification, PA = corrected classified sample number / total sample number in test sample.

**Change in manuscript:**

We explained the abbreviations in the titles of the corresponding tables.

L245: "Grassland is …" , from Table 3, they are shrubland and forest

**Response s30:**

It was concluded from the row dimension with the user's accuracy. But as for the producer's accuracy, it is as what you said.

L248: samples

**Response s31:**

Many thanks.

**Change in manuscript:**

We have corrected the word.

L255: these are regions with intensive LCC

**Response s32:**

Some regions such as Africa show relatively intensive LCC. There may be more mosaic pixels in these places in Africa, which may also lead to high uncertainty

For other regions with relatively high uncertainty, their locations are close to the continent edge which may be one of the reasons. The uncertainty map was reported based on the interpolation of test samples, the uncertainty values near the edges where test samples are rarely distributed would be affected to some degree.

L259: "variation curves" temporal changes

**Response s33:**

Thank you very much for your kindness.

**Change in manuscript:**

We have changed the phrase.

L260: Why so strong forest increase from 2006-2008? is it real?

**Response s34:**

We think it should be carefully treated. Since we do not have sufficient reference data, we cannot be sure if this is real or artifacts. The fluctuations in the curves can be seen as one of the representations of the uncertainty using coarse-resolution remotely sensed data.

L262: what about cropland? why so high variations, especially in 1994, 1999?

**Response s35:**

Cropland showed a slightly increasing trend, but not significantly. The high variations also reflect some kind of uncertainty introduced by the input data.

L263: Fig. 9, explain the meaning of your boxplot, mean, median, IQR, 90%, max, min? Why use the ratio, instead of total change area which is more straightforward?

**Response s36:**

The box extends from the first (lower) quartile (Q1) to third (upper) quartile (Q3) values of the data, with a line indicating the median. The whiskers extend from the box to show the range of the data. The upper whisker extends to the last datum less than Q3 + 1.5 * IQR, and the lower whisker extends to the first datum greater than Q1 - 1.5 * IQR. Flier points are those past the end of the whiskers. We wanted to use the change ratio to better reflect that how much percentage of global land cover changed in one year exactly like other studies (Friedl et al., 2010;Sulla-Menashe et al., 2019).

**Change in manuscript:**

We have added the corresponding introduction.

L263: "different time periods" the gross change each year or on the difference between the first and the last year in each period?

**Response s37:**

The annual ratio of the global land cover change area to the global total terrestrial area is plotted in

Fig. 9, but in a form of boxplot organized in a 5-year interval (a) and 10-year interval (b).

**Change in manuscript:**

We have revised our description.

L267: It's interesting to see a very likely decreasing trend of total LCC area.

**Response s38:**

Yes, and it was what our results showed.

L270: Fig. 10: The text and subplots in the figure is too small to read. I would suggest to only show the main land cover types and put the others to SI

**Response s39:**

Thanks for your suggestion.

**Change in manuscript:**

We have moved the low percentage classes such as shrubland, tundra and snow/ice to the supplementary information part.

L271: why only significant LCC? is it really necessary? why not just sum all?

**Response s40:**

In our opinion, the statistical test is necessary to lower the uncertainty in the long time-series land cover mapping results, especially for 0.05° such coarse resolution data.

L287: Table 5-10: too detailed, may put into SI and merge these results in a plot with different subplots

**Response s41:**

Thank you for the advice.

**Change in manuscript:**

We have reorganized the tables and put some into the supplementary information part.

L299: In In

**Response s42:**

Thanks for your correction.

**Change in manuscript:**

We have deleted the extra word.

L304-308: see my comments on greening above

**Response s43:**

Thanks again.

**Change in manuscript:**

We have deleted the corresponding part.

L309: need to note the high uncertainty from Fig. 7

**Response s44:**

Thank you for your valuable comment.

**Change in manuscript:**

We have added the note in the manuscript.

L310-311: greening again

**Response s45:**

Thanks again.

**Change in manuscript:**

We have deleted the corresponding part.

L313: Look at Fig. 10 and 11, significant grassland changed to forest in your dataset in the high latitudes

**Response s46:**

Yes, it is.

L316-317: why barren land decrease implies the desertification effects?

**Response s47:**

We may not expressed it clearly. What we mean was the management efforts to the desertification.

**Change in manuscript:**

We have updated the description.

L321: what is a coupling effect? non-relevant sentence

**Response s48:**

Thank you for the comment. What we mean was that natural and human factors usually had a significant joint effect on land cover change. Both aspects contribute to making a difference.

**Change in manuscript:**

We have revised the sentence.

L324 and all below: referring a or b when you report something. Why no explanations on the transitions from grassland to forest, which is the most obvious pattern in your figure

**Response s49:**

Thanks for your suggestion. This may be related to the shortcomings of the hard classification we adopted. As you pointed out above, the forest may become denser and the land cover class may change. But the interpretations of these phenomena are not the main focus of this paper.

L338: too strong statement. surface greening is not something that you can directly interpreted from LCC.

**Response s50:**

Thanks for your comment.

**Change in manuscript:**

We have changed the word.

L345: "natural vegetation" managed forest or pasture are not natural vegetation

**Response s51:**

Thank you for the comment. We used the wrong word.

**Change in manuscript:**

We have deleted the word "natural".

L346: how about reforestation?

**Response s52:**

Yes, human activities also include reforestation. The focus of this paper is still on data product introduction and evaluation, we will weaken this part of the introduction.

**Change in manuscript:**

We have revised our description to avoid the ambiguity.

L355: shy subtropical mountain system is also high?

**Response s53:**

Figure 6 shows the division of eco-regions from FAO, where regions in the orange color belong to subtropical mountain system. Referring to Fig. 5, they overlaps some regions with a relatively high human impact level, such as Spain, central China, east America and South Africa. These regions may bias the overall results.

[Figure]

Figure 6 Subtropical mountain system eco-regions from FAO.

L363: Fig. 15 is very misleading with only >0 and <0. Why not give gradient of change?

**Response s54:**

Thank you very much for your suggestion. Here, we do so because we want to more intuitively reflect the information about where gain or loss occurred.

L365 and below: again, give the subplot title when you describing the results.

**Response s55:**

Thanks again.

**Change in manuscript:**

We have added the information as you suggested.

L381: Do you have evidence that global warming will increase vegetation in tropics?!

**Response s56:**

This part of the analysis is not the focus of this paper.

L383-384: Oil palm plantations are forest or crop in your classification system? I am not sure whether you can distinguish them!

**Response s55:**

We are sorry for it. In our classification system, oil palm plantations are forest.

**Change in manuscript:**

We have adjusted the sentence.

L399: Is that partly why you detected forest increase at the expense of grassland?

**Response s57:**

We are afraid not. Mongolia and Inner Mongolia of China mostly belong to semi-arid regions. The land cover types there should be grassland or barren land. It should have nothing to do with the forest.

L421-423: yes, this is the main defect of this product.

**Response s58:**

Yes. This is also one of the common problems of hard classification.

L435: This is definitely something that has to be done in this work.

**Response s59:**

Thank you for your advice. In order to specifically evaluate the magnitude of the errors introduced by our training samples, we randomly selected 500 samples from the training samples for manual interpretation and evaluation, and the assessment accuracy was 92.26%. It shows that the training samples we generate this way are sufficient for our data production.

**Change in manuscript:**

L441: what about the heterogeneous pixels?

**Response s60:**

We have added new samples for comprehensive independent accuracy assessment, where heterogeneous samples are also included.

L460: NDVI and LAI increase not equal to forest increase

**Response s61:**

Indeed it is. NDVI and LAI are features that help forest classification, and we are not strict in saying so here.

**Change in manuscript:**

We have updated our words.

L455-463: not helping but expose the weakness of the product

**Response s62:**

Thanks for your comment.

**Change in manuscript:**

We have deleted the corresponding part.

L464-465: This contradicts that you said forest loss in SE Asia is due to oil palm plantations

**Response s63:**

We are sorry for it. In our classification system, oil palm plantations are forests.

L466-467: need more explanations to justify the reasons for doing this.

**Response s64:**

As mentioned above, this is the result of our trade-off. There are too many uncertain factors in remote sensing. In contrast, suppressing some real fluctuations in LCC, and performing post-processing in the time dimension can make data products more reliable, less uncertain and less noisy. And the accuracy improvement brought by change detection illustrates the effectiveness of doing so.

---

## Author Comment (AC2) · 18 Jan 2020

We thank the reviewer for the comments and thoughtful review. Please find our detailed response along with the suggested changes to our manuscript below.

General comment:

By fusing multiple existing geo-spatial datasets, the main work of this manuscript is to generate an annual dynamic product (spatial resolution: 0.5°) addressing seven kinds of land-covers (i.e., cropland, forest, grassland, shrub-land, tundra, barren land and snow/ice) from 1982 to 2015. With eye on the current existing datasets (i.e., from the perspective of classification system, period of time, and spatial/temporal resolution) the contribution is quite limited. In view of the rationality of technique and accuracy assessment, current version calls for serious revision before publication. In view of the analysis conducted on the dynamic map, rare novel findings can be captured.

**Response 1**:

Thanks for the comment. First of all, this is a paper describing a unique data product. It is not 0.5 degrees in resolution but 5 km. Since it is about land cover data product, it is not our attention to make novel discoveries. The purpose here is mainly to present a data set that does not exist anywhere before, for its annual frequency, 34 years long duration and high accuracy. The classification system does cover more than 90% of the land area. We did not include water, wetland, and impervious areas because wetland is extremely dynamic (more frequent than the yearly scale), water excluded from the input data source, and impervious areas already processed using more accurate source of data (e.g., annual Global Artificial Impervious Area maps, (Gong et al., 2020)). The accuracy assessment has been further improved using additional collection of test samples. We also compared our results with other data products and found that our results are superior.

Specific comments:

There are several global datasets with more rigorous production process have existed. 1) The 1992-2018 annual 300m global land-cover data (https://www.esa-landcover-cci.org/?q=node/197 ) with more detailed classification scheme have been released. Since the proposed product has no accuracy assessment on the annual maps from 1982-1991, it cannot be argued that the proposed work have longer period of time.

**Response 2**:

Thanks for your comment. We agree that ESA-CCI products have higher spatial resolution and more detailed classes. However, products with different resolution have different application purposes. In many studies, it is only necessary to use coarse-resolution land cover data, such as our 0.05 ° data, which can be used in Earth system modeling.

For Earth system modeling purposes, the 10 land cover classes mentioned in our response at the beginning are sufficient. Among the ten classes, except for wetland, impervious area, and water that

occupy less than 10% of the entire land area on Earth. In the meantime, water and impervious areas can be individually obtained. Wetland is highly dynamic requiring additional types of remotely sensed data. Considering the separability and identifiability of the land cover classes under the 5 km spatial resolution, we adopted a classification system of 7 classes.

In this revision, we collected new independent test samples and performed accuracy assessment for the period of 1982-1991. In addition, we have compared our products with ESA-CCI and MODIS-based land cover data products and FAOSTAT data. The results show that our products have good reliability.

Specifically, we collected 2431 randomly distributed 5km sample points in different years around the world. According to the majority principle, we manually interpreted the land cover class of each sample as an independent test sample. Besides, in order to verify the accuracy of the change detection method, we also compared the classification accuracy before and after the change detection. The temporal distribution of the newly collected test samples is shown in Fig. 1, and the geographical distribution is shown in Fig. 2.

[Figure]

Figure 1: The temporal distribution of the newly collected test sample.

[Figure]

Figure 2: The geographical distribution of random test sample.

[revised manuscript text omitted]

**Change in manuscript:**

We have added the new accuracy assessment results and data inter-comparison results to help show the reliability and effectiveness of GLASS-GLC.

2) The annual VCF products from 1982- 2016 have the same spatial resolution, and very similar classification scheme with the proposed work (from 1982-2015). Although the VCF product is missing in 1994 and 2000, the proposed work just directly use the data-source around the adjacent year, which cannot be viewed as a noticeable contribution. Meanwhile, since the proposed work also introduce VCF in the supervised classification, the analysis on the dynamic map is somewhat similar to this existed study (Song et al 2018a) , but more superficial.

**Response 3**:

Thank you for your comment. VCF is a quantitative variable. VCF data products mainly reflect vegetation cover information. Our land cover classes include multiple types of nominal variables. Again VCF and land cover information have different purposes of applications.

Here, we introduce VCF as a priori information to assist in land cover classification. VCF data is missing for two years, but this will not greatly affect the classification results. The auxiliary or supplementary data for classification and interpretation do not need to be perfect. They do not need to be in the same period or at the same resolution. As long as there is supplementary information, it will work, such as in the four-dimensional variational data assimilation.

For the analysis part of the land cover classification, the results are similar to those obtained from the analysis of VCF, which also confirm the objectivity and correctness of VCF analysis. But it is worth pointing out that our products can analyze many more detailed classes, so we can also draw some different conclusions.

Considering that the type of this paper is a data paper, our main focus is on the description of the production methods and quality control of data products, and the comparison and analysis of data quality and accuracy. More in-depth LCC analysis is out of the scope of this study.

Technical corrections

1. It is ridiculous to produce training and test set from a same product and in a same manner. In addition, it is unacceptable to conclude the applicability of the long-time period product by assessing the accuracy on only the 2015 land-cover mapping result.

**Response 4**:

Thank you for your comment. There may be some flaws in the way we evaluate accuracy.

Taking this into consideration, in addition to the accuracy assessment of samples taken from the FROM-GLC_v2 product, samples from FLUXNET site data are also given for independent accuracy assessment. The assessment results are shown in Table 3. The overall accuracy of GLASS-GLC reached 82.10% in 2015.

In addition, as described in response 2 above, we conducted a new independent sample test (OA=82.81%) and a comparison of multiple products (land cover products from MODIS and ESA-CCI, and FAOSTAT data), which also proved the reliability of our products.

**Change in manuscript:**

We have added the new accuracy assessment results and data inter-comparison results to help show the reliability and effectiveness of GLASS-GLC.

2. How to project the 30m FROM-GLC_v2 to mapping scale? How to deal with the mixed sample?

**Response 5**:

Thank you for your question. As the paper says, we projected the results of FROM-GLC_v2 according to the principle of majority. That is, the land cover class that accounts for the largest proportion in each grid is used as the land cover class label under the 0.05 ° grid. Generating coarse-resolution samples from high-resolution products as such is actually a common practice (Wang et al., 2016;DeFries et al., 1998).

For mixed samples, we also use the majority principle to give labels. Although percentage information is more suitable for dealing with mixed pixels, our goal here is hard classification, and we cannot avoid only doing so. This is also a problem that arises in hard land classification studies.

Considering the cost constraints, we have adopted this method of generating new samples even though it will bring some errors when producing coarse resolution samples from FROM-GLC_v2. However, the "stable classification with limited sample" theory (Gong et al., 2019) supports our approach to some extent. The theory shows that under its experimental conditions, even if 20% of the wrong samples are introduced, the classification accuracy is reduced by 1%, and it can still be stable (Fig. 7).

[Figure]

Figure 7: Sample robustness to size reduction and errors in sample. a. As sample size increases, the accuracy quickly reaches a plateau. b. As the impurity percentage of sample increases the accuracy decreases. In both cases, the 1000 times random drawing of sample points produced very stable overall classification accuracies with most standard deviations much lower than 0.5%. (Gong et al., 2019)

The newly added results of accuracy assessment have also confirmed that the samples produced in this way can meet the production needs.

3. There is no sample accuracy assessment on the produced training sample set. Please note that the accuracy of the FROM-GLC_v2 is not high enough to work as training sample.

**Response 6**:

Thank you for your point. The classification accuracy of FROM-GLC_v2 will surely have some impact on our results. However, FROM-GLC_v2 has been published, and it has a detailed accuracy assessment, with an OA of 73.13%. There is some complicated relationship for the use of higher resolution land cover data in producing lower-resolution land-cover products. Since there is a scaling down which requires aggregation of high-resolution land cover results. This often acts as an averaging effect that improves the accuracy in the area to some extent. Even if there is no accuracy increase during the scaling down process, the 73% accuracy would not cause a large accuracy decrease as can be seen from the figure in the right-hand side of Fig. 7.

To specifically evaluate the magnitude of the errors introduced by our training samples, we randomly selected 500 samples from the training samples for manual interpretation and evaluation, and the assessment accuracy was 92.26%. It shows that the training samples we generate this way are sufficient for our data production.

**Change in manuscript:**

We have added the accuracy assessment results on training samples.

4. When mapping the land-covers decades year ago, the suitability of the samples collected (mainly from 2013-2015) should be evaluated.

**Response 7**:

Thanks for your advice. We agree that, in the early years, the percent of land cover change may be relatively large. However, global land cover will not change by more than a few percents for decades. And these changes are primarily in urban and urban-rural fringe areas. The outdatedness of samples will not affect much of our accuracy assessment.

Concerning the reliability of sample migration, the "stable classification with limited sample" theory is specifically discussed (Gong et al., 2019).

In this study, the concept of a stable classification is defined. They use this concept to approximately determine how much reduction in training sample and how much land cover change or image interpretation error can be acceptable. If the mean accuracy of multiple runs of a classifier trained with a random drawing of a certain percentage of sample points from the total sample is within 1% of what can be achieved with the total sample set, we regard the obtained classification result "stable". The 1% threshold is empirically chosen based on the fact that a loss of overall accuracy in 1% shall not significantly impact the application of a global land cover map.

Tens of millions of experiments suggest that it is possible to use 60% fewer sample points and even the land cover changed by 20% or the training sample contains 20% errors, we are still able to achieve "stable" classification with the random forest classifier in global land cover mapping. This conclusion well supports the effectiveness of our sample transfer method. Even for decades, it is

difficult for global land cover to change by more than 20%. Therefore, the proportion of error samples we introduced in the early years will not exceed 20%, and the classification results are still reliable and effective.

Another recent study (Huang et al., 2020) also devoted to migrating training samples to early years. They developed an automatic training sample migration method, which can successfully migrate training samples in 2015 to 2000. These studies prove the effectiveness of sample migration and provide potential solutions to resolve the problem of lack of training samples for dynamic global land cover mapping efforts.

Besides, to verify the temporal accuracy of our products, as mentioned above, we have independently collected test samples from different years and tested the accuracy of our products, with an accuracy of 82.81%. What's more, the inter-comparison results with other data have also confirmed the validity of our data using the 2015 sample for many years.

---

## Author Response (AR2)

**Response to Editor**

Dear Authors and Colleagues

thanks for the authors for the replies to the reviews of your paper and for the revision. We require moderate revision of the data publication and the manuscript.

I look forward to your final data publication and ESSD manuscript,
Best wishes, Birgit Heim

Dear Editor,

We appreciate your prompt comments.
Please find our detailed responses along with the suggested changes below.
Thanks again.

Best wishes,

Han Liu
On behalf of all authors

In general please let check the language, there are still minor issues.

i) please change 'feature collection' and 'feature' throughout the text, title, data description – as it is a more technical term from GIS and machine learning. for example, you could use 'input data collection'

**Response 1**:

Thanks for your advice.

**Change in manuscript:**

We have checked and updated the expression and converted "feature collection" into 'input data collection' in the manuscript.

ii) table 1 shows the 7 classes level-1 but also level-2 classes. It does not become clear to the reader if table 1 describes the GLASS GLC product. The GLASS GLC product contains information on level 1 only. Better to show this table as table in results with the 7 classes as GLASS GLC classes. in the other column that is now named level 2 and contains the subclasses the title should not be 'level 2' put here better a reference of the other Land Cover product that refers to these subclasses

**Response 2**:

Thanks for your suggestion.

**Change in manuscript:**

We have revised the words 'level 1' and 'level 2' and provided a clearer description in the whole manuscript.

**Table 1: Classification system, with 7 land cover classes.**

| Class | Subclass with reference to (Li et al., 2017) | Description |
|---|---|---|
| Cropland | Rice paddy
Greenhouse
Other farmland
Orchard
Bare farmland | |
| Forest | Broadleaf, leaf-on
Broadleaf, leaf-off
Needle-leaf, leaf-on
Needle-leaf, leaf-off
Mixed leaf type, leaf-on
Mixed leaf type, leaf-off | Tree cover≥10%;
Height>5m;
For mixed leaf, neither coniferous nor broadleaf types exceed 60% |
| Grassland | Pasture, leaf-on
Natural grassland, leaf-on
Grassland, leaf-off | Canopy cover≥20% |
| Shrubland | Shrub cover, leaf-on
Shrub cover, leaf-off | Canopy cover≥20%;
Height<5m |
| Tundra | Shrub and brush tundra
Herbaceous tundra | |
| Barren land | Barren land | Vegetation cover<10% |
| Snow/Ice | Snow
Ice | |

table 2 shows the input data in your processing chain and fits well in the subchapter describing the input data collection. Please explain all abbreviations in the figure captions. Please add the information what is static and what is dynamic and with which temporal resolution.

**Response 3**:

Thanks for your comment.

**Change in manuscript:**

We added explanations to all abbreviations and temporal information in Table 2 as follows.

**Table 2: The explanatory table of the constructed input data collection\***

| Product | Band | Input | Number |
|---|---|---|---|
| GLASS CDR, 0.05 °, 8-day, dynamic, 1982-2015 | NDVI LAI FAPAR ET GPP | Percentiles [0, 10, 25, 50, 75, 90, 100] of all 9 bands | 63 |
| | BBE ABD_WSA_VIS ABD_BSA_NIR ABD_WSA_shortwave | Mean, standard derivation of NDVI between adjacent two percentiles of NDVI | 12 |
| VCF, 0.05 °, yearly, dynamic, 1982-2015 | TC SV BG | TC SV BG | 3 |
| GMTED2010, 7.5 s, static, 2010 | Elevation | Mean slope in each 0.05 ° pixel | 1 |
| Location, static | Latitude, longitude | Center latitude and longitude of each 0.05 ° pixel | 2 |
| **Total** | | | **81** |

\* GLASS CDR represents the global land surface satellite climate data records. VCF stands for vegetation continuous fields, and GMTED2010 refers to global multi-resolution terrain elevation data of 2010. NDVI, LAI, FAPAR, ET, GPP, and BBE are abbreviations for normalized difference vegetation index, leaf area index, fraction of absorbed photosynthetically active radiation, evapotranspiration, gross primary production, broadband emissivity, respectively. ABD_WSA_VIS, ABD_BSA_NIR, and ABD_WSA_shortwave represent white-sky albedo in visible band, near infrared band, and shortwave band, respectively. TC, SV, and BG stand for tree canopy, short vegetation, and bare ground cover, respectively.

PANGAEA data publication, required changes
i) PANGAEA abstract
It is important that the PANGAEA data publication stands for its own with all information available. Please add to the abstract the detailed information about the data set: geographic and temporal coverage, spatial and temporal resolution, projection, geotiff format. Please list the seven land cover classes in the PANGAEA abstract. Please add also the information in the abstract that links to your ESSD publication.

**Response 4:**

Thanks for your reminding.

**Change in manuscript:**

We have added the information you mentioned in the abstract of the data set as follows.

Land cover is the physical evidence on the surface of the Earth. As the cause and result of global environmental change, land cover change (LCC) influences the global energy balance and biogeochemical cycles. Continuous and dynamic monitoring of global LC is urgently needed. Effective monitoring and comprehensive analysis of LCC at the global scale are rare. With the latest version of GLASS (The Global Land Surface Satellite) CDRs (Climate Data Records) from 1982 to 2015, we built the first record of 34-year long annual dynamics of global land cover (GLASS-GLC) at 5 km resolution using the Google Earth Engine (GEE) platform. Compared to earlier global LC products, GLASS-GLC is characterized by high consistency, more detailed, and longer temporal coverage. The average overall accuracy for the 34 years each with 7 classes, including cropland, forest, grassland, shrubland, tundra, barren land, and snow/ice, is 82.81 % based on 2431 test sample units. We implemented a systematic uncertainty analysis and carried out a comprehensive spatiotemporal pattern analysis. Significant changes at various scales were found, including barren land loss and cropland gain in the tropics, forest gain in northern hemisphere and grassland loss in Asia, etc. A global quantitative analysis of human factors showed that the average human impact level in areas with significant LCC was about 25.49 %. The anthropogenic influence has a strong correlation with the noticeable vegetation gain, especially for forest. Based on GLASS-GLC, we can conduct long-term LCC analysis, improve our understanding of global environmental change, and mitigate its negative impact. GLASS-GLC will be further applied in Earth system modeling to facilitate research on global carbon and water cycling, vegetation dynamics, and climate change. This GLASS-GLC data set is related to the paper at https://www.earth-syst-sci-data-discuss.net/essd-2019-23. It consists of one readme file and 34 GeoTIFF files of annual 5 km global maps from 1982 to 2015 in a WGS 84 projection.

ii) PANGAEA data publication
GLASS GLC should be a part in the file naming. Currently, the files in your product are named cls_allbands_adj19_mask_1982, etc., referring to internal processing issues? Please change to a more user-friendly product name. For example you can include that the product has 7 classes: GLASS-GLC_7classes_year or similar.
Please address PANGAEA for exchanging your data collection. This is feasible as the DOI provided by PANGAEA during the review of your ESSD manuscript and data set is a review link with the possibility to exchange the data publication until your ESSD manuscript will be finally published.

**Response 5**:

Thanks for your advice.

**Change in manuscript:**

We have changed the file name of data set in the new form 'GLASS-GLC_7classes_year' as you suggested. And the updated data set is placed at https://doi.pangaea.de/10.1594/PANGAEA.913496.

iii) Readme file

please add a readme file to the PANGAEA data collection instead of the word document 'labelinstruction.doc'. Please include in the readme file a short data description, the geographical and temporal coverage, the projection and the spatial resolution in addition to the overview on the classes.

**Response 6**:

Thanks for your suggestion.

**Change in manuscript:**

We have updated the readme file as follows.

The GLASS-GLC data set is the first record of 34-year long annual dynamics of global land cover spanning from 1982 to 2015 at 5 km resolution. It was built with the latest version of GLASS (The Global Land Surface Satellite) CDRs (Climate Data Records) and generated on the Google Earth Engine (GEE) platform. The average overall accuracy for the 34 years each with 7 classes, including cropland, forest, grassland, shrubland, tundra, barren land, and snow/ice, is 82.81 %.

The annual global land cover map (5 km) is presented in a GeoTIFF file format named in the form of 'GLASS-GLC_7classes_year' with a WGS 84 projection. The relationship between the labels in the files and the 7 land cover classes is shown in the following table.

Table 3 Classification system, with 7 land cover classes.

| Label | Class | Subclass with reference to (Li et al., 2017) | Description |
|-------|-------|----------------------------------------------|-------------|
| 10 | Cropland | Rice paddy | |
| | | Greenhouse | |
| | | Other farmland | |
| | | Orchard | |
| | | Bare farmland | |
| 20 | Forest | Broadleaf, leaf-on | Tree cover≥10%; |
| | | Broadleaf, leaf-off | Height>5m; |
| | | Needle-leaf, leaf-on | For mixed leaf, neither |
| | | Needle-leaf, leaf-off | coniferous nor broadleaf |
| | | Mixed leaf type, leaf-on | types exceed 60% |
| | | Mixed leaf type, leaf-off | |

| 30 | Grassland | Pasture, leaf-on
Natural grassland, leaf-on
Grassland, leaf-off | Canopy cover≥20% |
|---|---|---|---|
| 40 | Shrubland | Shrub cover, leaf-on
Shrub cover, leaf-off | Canopy cover≥20%;
Height<5m |
| 70 | Tundra | Shrub and brush tundra
Herbaceous tundra | |
| 90 | Barren land | Barren land | Vegetation cover<10% |
| 100 | Snow/Ice | Snow
Ice | |
| 0 | No data | | |

For more details, please refer to the paper at https://www.earth-syst-sci-data-discuss.net/essd-2019-23. When you use the data set, please also cite this paper as follows:

*Liu, Han, Peng Gong, Jie Wang, Nicholas Clinton, Yuqi Bai, and Shunlin Liang. "Annual Dynamics of Global Land Cover and its Long-term Changes from 1982 to 2015." Earth Syst. Sci. Data Discuss 2019 (2019): 1-54.*

Contact information
Han Liu (liuhan18@mails.tsinghua.edu.cn)
Peng Gong (penggong@tsinghua.edu.cn)

Abstract
The starting sentence of the abstract ' Land cover is an important terrestrial variable for understanding the interaction between human activities and global change ' seems to be misleading as the interaction between human activities and global change are a part of your study only, but not the focus and the main product. Please change the focus of the starting sentence of your abstract.

**Response 7**:

Thanks for your advice.

**Change in manuscript:**

We have changed the sentence to 'Land cover is the physical evidence on the surface of the Earth'.

L21: please remove the statement 'GLASS-GLC is superior in the accuracy and the temporal range '- the group of existing land cover products is diverse in temporal, spatial and thematic resolution and produced for different user groups and applications.

**Response 8**:

Thanks for your suggestion.

**Change in manuscript:**

We have removed the sentence.

Introduction
p.3 L95 to 101 please reorder the land cover product description, put the spatial resolution in brackets at the end of the description and the characteristic of the product, if it is static or dynamic: e.g., instead of 1 km University of Maryland xxx land cover map change to land cover product xxx ( 1 km pixel resolution)

**Response 9**:

Thanks for your suggestion.

**Change in manuscript:**

We have revised all the descriptions in the corresponding position.

Some examples include Finer Resolution Observation and Monitoring of Global Land Cover product (FROM-GLC) in 2010, 2015 and 2017 (30 m and 10 m) (Gong et al., 2013;Gong et al., 2019), European Space Agency Climate Change Initiative land cover data (ESA-CCI) from 1992 to 2015 (300 m) (http://maps.elie.ucl.ac.be/CCI/viewer/index.php, last access: 20 November 2018), MODIS Land Cover Type series products (MLCT) from 2001 to 2016 (500 m) (Friedl et al., 2010;Sulla-Menashe et al., 2019), International Geosphere-Biosphere Programme Data and Information System Cover map (IGBP-DISCover) (1 km) circa 1992 (Loveland et al., 2000), University of Maryland (UMD) land cover map circa 1992 (1 km) (Hansen et al., 2000), Global Land Cover 2000 map (GLC2000) (1 km) (Bartholome and Belward, 2005).

p.3 L108-109 Landsat data …. 'higher' sounds not specific -> please add 'spatial' to resolution and add the information of 30 m pixel resolution. Please refrain the issue with cloud cover in explaining the problem of cloud cover and Landsat with the reduced temporal resolution (as also AVHRR and MODIS have the cloud cover problem but due to much higher temporal resolution the cloud cover is less a problem compared to Landsat). Would not state the large data volume nowadays as such a problem, or you would need to state that you refer to your case to produce land cover maps with global coverage over tens of years

**Response 10**:

Thanks for your helpful comment.

**Change in manuscript:**

We have changed the sentences as you suggested.

Landsat data has a higher spatial resolution of 30 m with some restrictions including more serious cloud contamination problem owing to a low temporal frequency, and data inconsistency problem caused by multiple generations of sensors (Gómez et al., 2016;Wulder et al., 2008;Xie et al., 2018).

Data and Methods

The subtitles should be more specific
2,4, feature collection, 2.5 classification and time consistency, titles not clear
2.8 statistical analyses, of what?, 2.9 LC conversion -> title not clear, do you mean change? – do 2.8 and 2.9 fit together?
2.10 human impact -> add a process

**Response 11**:

Thanks for your advice.

**Change in manuscript:**

We have changed the subtitles as follows.

2.4 Input data collection
2.5 Classification method and temporal consistency check
2.8 Statistical analysis of LCC
The initial section of 2.8 described the statistical method to estimate significant land cover change patterns and trends, and the initial section of 2.9 focused on the extraction method to explore the conversion or change relationships among different land cover classes. Considering the two parts are both about methods to analyze land cover change, we merged the two sections into one.
2.9 Human impact process

table 2 The GLASS CDR products seem to account to nine products – why are the percentiles calculated for 10 bands?

**Response 12**:

Thanks for your kind correction.

**Change in manuscript:**

We have corrected it.

p.5 L178 DEM as the DEM input does not represent a single year but represents a multi-year merged product, please refrain the sentence 'using data of a single year'

**Response 13**:

Thanks for reminding us.

**Change in manuscript:**

We have changed the description as 'As the terrain is relatively stable over years, using the data as input for multiple years is plausible'.

results
p.11 L454 the evaluation of GLASS-GLC shows good results. However, the phrase that GLASS CDR products are superior is a much too strong statement. Please remove the statement

**Response 14**:

Thanks for your suggestion.

**Change in manuscript:**

We have removed the word.

the ice and snow class in the GLASS-GLC yearly product represents perennial ice and snow cover, please specify it in the text as perennial ice or snow. You would not need to add it to tables or figures but add the description in the text, and if you describe the change, you could specify where the reduction from perennial ice and snow takes place and where which land cover class replaces perennial ice and snow.

**Response 15**:

Thanks for your suggestion.

**Change in manuscript:**

We have added the specification and descriptions in the text.

Figure 8: please exchange MODIS based with the names of the Land Cover products

**Response 16**:

Thanks for your advice.

**Change in manuscript:**

We have changed 'MODIS-based' to the product name 'MLCT' in the manuscript.

[revised manuscript text omitted]

---

## Author Response (AR3)

Dear Authors and Colleagues

thanks for the authors for the replies to the reviews of your paper and for the revision. We require minor edits in the manuscript and tables.

I look forward to your final data publication and ESSD manuscript,
Best wishes, Birgit Heim

Dear Editor,

We appreciate your great advice and the whole excellent work and efforts for our paper.
We have given our detailed responses along with the suggested changes to our manuscript below.
Thanks again.

Best wishes,
Han Liu
On behalf of all authors

Requirements

i) please let check the language, there are still minor issues

**Response 1**:

Thanks for your advice.

**Change in manuscript:**

We have polished our language with a native English consultant.

ii) please check carefully to refer to the PANGAEA DO 913496 data only https://doi.pangaea.de/10.1594/PANGAEA.913496
please change the data set DOI in chapter 5 and references accordingly

**Response 2**:

Thanks for your kindly reminding.

**Change in manuscript:**

We have updated the DOI information of our dataset

(https://doi.pangaea.de/10.1594/PANGAEA.913496) in the whole manuscript.

iii) please cite your sources accordingly in chapter 2.1 data sources, in table 2 (you could add an additional column) and in chapter 5 data availability

- e.g., GLASS ad citation Liang in chapter 2.1, table 2

- important: please cite VCF product type and product guide in chapter 2.1 and table 2 and references:
VCF5kyrv001 MEaSUREs Vegetation Continuous Fields (VCF) Yearly Global 0.05 Deg,
MEaSURES Vegetation Continuous Fields ESDR Algorithm Theoretical Basis Document (ATBD)
You could abbreviate to VCF5kyry001 in table 2

- GMTED2010, please cite USGS: USGS Report 2011–1073 Global Multi-resolution Terrain Elevation Data 2010 (GMTED2010) in chapter 2.1, table 2, and reference
Important: the global GMTED2010 dataset was published in 2010, however it does not represent the state of elevation data in 2010, as many of its DEM sources are at least one decade older, the global GMTED2010 dataset is a multi-year synthesis data set.
In table 2
GMTED2010, 7.5 arc seconds, static (please remove here 2010 as this could lead to the misunderstanding that the DEM represents the status of 2010 elevation)

**Response 3**:

Thanks for your suggestion and correction.

**Change in manuscript:**

We have checked and updated all the data citations in chapter 2.1, table 2, chapter 5, and reference.

[revised manuscript text omitted]